# GT-rich promoters can drive RNA pol II transcription and deposition of H2A.Z in African trypanosomes

Carolin Wedel[1] ID, Konrad U Förstner[2] ID, Ramona Derr[1] & T Nicolai Siegel[1,3,4,*] ID

## Abstract

Genome-wide transcription studies are revealing an increasing number of "dispersed promoters" that, unlike "focused promoters", lack well-conserved sequence motifs and tight regulation. Dispersed promoters are nevertheless marked by well-defined chromatin structures, suggesting that specific sequence elements must exist in these unregulated promoters. Here, we have analyzed regions of transcription initiation in the eukaryotic parasite *Trypanosoma brucei*, in which RNA polymerase II transcription initiation occurs over broad regions without distinct promoter motifs and lacks regulation. Using a combination of site-specific and genome-wide assays, we identified GT-rich promoters that can drive transcription and promote the targeted deposition of the histone variant H2A.Z in a genomic context-dependent manner. In addition, upon mapping nucleosome occupancy at high resolution, we find nucleosome positioning to correlate with RNA pol II enrichment and gene expression, pointing to a role in RNA maturation. Nucleosome positioning may thus represent a previously unrecognized layer of gene regulation in trypanosomes. Our findings show that even highly dispersed, unregulated promoters contain specific DNA elements that are able to induce transcription and changes in chromatin structure.

**Keywords** core promoter; histone variant; nucleosome occupancy; *Trypanosoma brucei*

**Subject Categories** Chromatin, Epigenetics, Genomics & Functional Genomics; Microbiology, Virology & Host Pathogen Interaction; Transcription

The EMBO Journal (2017) 36: 2581–2594

## Introduction

Transcription initiation involves the assembly of an elaborate complex of transcription factors to target the polymerase to the correct genomic locus. Two key features of this process are a core promoter, which functions as a platform on which the transcription machinery assembles, and a chromatin structure that allows the transcription machinery to access the promoter DNA (Kadonaga, 2012). Typically, the core promoter is defined as the minimal DNA sequence necessary and sufficient to direct the accurate initiation of transcription by an RNA polymerase (RNA pol) (Roeder, 1996). Core promoters commonly comprise specific DNA elements that confer distinct properties to the promoter; for example, they serve as binding sites for transcription factors (Roeder, 1996) or are involved in defining a specific chromatin structures that leave the DNA depleted of nucleosomes (Raisner *et al*, 2005; Yuan *et al*, 2005; Mavrich *et al*, 2008a).

While most eukaryotic DNA is occupied by nucleosomes, genome-wide nucleosome occupancy maps that have been generated for more than 30 organisms have revealed a consistent pattern: In all organisms studied thus far, promoters and other regulatory elements are depleted of nucleosomes (Hughes & Rando, 2014). In addition, there is an enrichment of the histone variant H2A.Z at the nucleosomes that flank the nucleosome-depleted regions (NDRs) (Albert *et al*, 2007; Barski *et al*, 2007; Mavrich *et al*, 2008b). Reports regarding the biological function of H2A.Z are conflicting, with some studies describing that incorporation of H2A.Z stabilizes the nucleosome and others describing a destabilizing effect (reviewed in Zlatanova & Thakar, 2008). Similarly, some studies report an activating role for H2A.Z on transcription, while others describe a repressive influence (reviewed in Marques *et al*, 2010).

The most important factor in the establishment of NDRs seems to be the DNA sequence (Struhl & Segal, 2013). Homopolymeric sequences such as poly(dA:dT) and poly(dG:dC) are intrinsically rigid, a property that negatively affects the ability of DNA to bend around histone octamers and thus strongly inhibits nucleosome formation (Suter *et al*, 2000; Segal & Widom, 2009). The mechanisms leading to the deposition of H2A.Z at promoters remain less well understood, but findings in yeast suggest that deposition may be dependent on the presence of NDRs (Raisner *et al*, 2005), which are recognized by the chromatin remodeling complexes SWR-C/

1 Research Center for Infectious Diseases, Universität Würzburg, Würzburg, Germany
2 Core Unit Systems Medicine, Universität Würzburg, Würzburg, Germany
3 Department of Veterinary Sciences, Experimental Parasitology, Ludwig-Maximilians-Universität München, München, Germany
4 Biomedical Center Munich, Physiological Chemistry, Ludwig-Maximilians-Universität München, Planegg-Martinsried, Germany
*Corresponding author. Tel: +49 89 2180 77098; E-mail: n.siegel@lmu.de

SWR1 (Yen *et al*, 2013). SWR-C/SWR1 had previously been shown to catalyze the exchange of H2A with H2A.Z (Mizuguchi *et al*, 2004).

Different genome-wide methods have identified a rapidly increasing number of promoters that lack well-defined sequence motifs and revealed two general classes of promoters: "focused" and "dispersed" (Carninci *et al*, 2006). Historically, the majority of research has been devoted to focused promoters where transcription initiation occurs from a single transcription start site (TSS) or a narrow cluster of TSSs that are associated with well-defined core promoter motifs, for example, a TATAA box. However, it is becoming increasingly clear that a large percentage of promoters (~70% in mammals, Saxonov *et al*, 2006) are dispersed, typically containing TSSs spread over a range of 50–100 bp, although regions as wide as 10 kb have been reported (Koch *et al*, 2011). Besides mammals, dispersed promoters have been identified in yeast (Zhang & Dietrich, 2005), *Arabidopsis thaliana* (Yamamoto *et al*, 2009), *Drosophila melanogaster* (Ni *et al*, 2010), *Xenopus laevis* (van Heeringen *et al*, 2011), *Leishmania major* (Martinez-Calvillo *et al*, 2003), and *Trypanosoma brucei* (Kolev *et al*, 2010). It is likely that more dispersed promoters will be discovered during similar genome-wide TSS studies in less well-studied organisms, which will help to elucidate the biological significance of this apparent promoter dichotomy.

Data from vertebrates and metazoans indicate that focused promoters drive transcription of highly regulated genes while dispersed promoters drive transcription of weakly regulated genes that lack well-defined sequence motifs (Sandelin *et al*, 2007; Lenhard *et al*, 2012). This correlation has raised questions about the role of promoter sequence motifs, for example, whether specific promoter motifs are required for transcription initiation of ubiquitously expressed genes or whether promoter motifs exist in organisms that do not regulate transcription initiation. In the latter case, it is possible that an "open" chromatin structure would permit transcription factors and polymerase to access the DNA and be sufficient for transcription initiation.

To elucidate the link between promoter structure and transcriptional regulation, we investigated the role of promoter sequence elements on transcription and histone variant deposition in an organism lacking transcriptional regulation, such as *T. brucei*. Unlike any eukaryotes studied thus far, this protozoan parasite belonging to the order of Trypanosomatida, which encompasses clinically important pathogens such as *L. major*, *Trypanosoma cruzi*, and *T. brucei*, appears to completely lack the ability to regulate transcription by RNA pol II (Clayton, 2002).

In these pathogens, RNA pol II-transcribed protein-coding genes are organized in long polycistronic transcription units (PTUs) with no clustering of functionally related genes (Berriman *et al*, 2005). Primary transcripts are co-transcriptionally processed into individual mRNAs by coupled *trans*-splicing and polyadenylation reactions (LeBowitz *et al*, 1993; Matthews *et al*, 1994). In *trans*-splicing, a 39-nt spliced-leader sequence containing the cap structure required for translation and possibly other aspects of mRNA function is added to the 5′-end of the 5′ UTR (Campbell *et al*, 1984; Kooter *et al*, 1984; Sutton & Boothroyd, 1986; Perry *et al*, 1987). Consistent with the notion that ubiquitously expressed genes lack well-defined promoter motifs, RNA pol II promoter motifs have been elusive in *T. brucei*, with the exception of the spliced-leader RNA promoter

(Das *et al*, 2005; Schimanski *et al*, 2005). Furthermore, RNA pol II transcription initiation in *T. brucei* can occur in the absence of identifiable promoter motifs (Marchetti *et al*, 1998; McAndrew *et al*, 1998) suggesting that a permissive chromatin structure may be sufficient for transcription initiation in this parasite. In agreement with this hypothesis, several genome-wide studies have found the boundaries of PTUs to be marked by distinct histone variants, such as H2A.Z, and histone modifications (Siegel *et al*, 2009; Thomas *et al*, 2009; Wright *et al*, 2010).

As seen in other organisms, H2A.Z is found at sites of RNA pol II transcription initiation in *T. brucei*. However, H2A.Z deposition is not restricted to narrow, well-defined sites upstream of individual genes; instead, H2A.Z is enriched across regions of ~10 kb in width, encompassing the first few genes of every PTU (Siegel *et al*, 2009). How H2A.Z is targeted to sites of transcription initiation in the absence of DNA motifs remains an enigma. In addition, high-throughput sequencing of primary transcripts suggests that RNA pol II transcription initiation occurs over broad genomic regions, matching sites of H2A.Z enrichment, rather than at well-defined TSSs (Kolev *et al*, 2010). We thus use the term transcription start regions (TSRs) rather than TSSs and define TSRs as regions enriched in H2A.Z compared to the rest of the genome. Nucleosome occupancy has not been systematically analyzed in trypanosomatids, but several studies find RNA pol I-transcribed genes to be depleted of nucleosomes (Figueiredo & Cross, 2010; Stanne & Rudenko, 2010).

In this study, we aimed to elucidate the features that define RNA pol II TSRs in *T. brucei* by determining the importance of DNA sequence elements in transcription initiation, histone variant recruitment, and nucleosome positioning.

We have generated for the first time genome-wide maps of nucleosome occupancy and RNA pol II enrichment in *T. brucei* as well as an improved dataset of TSSs. Using these datasets and data from other epigenome studies, we devised a dual-luciferase reporter assay to systematically test the ability of specific DNA elements to initiate transcription. Our study identified a GT-rich promoter motif, able to initiate transcription and to aid in the targeted deposition of the histone variant H2A.Z. In addition, we find that TSRs are more sensitive to micrococcal nuclease (MNase) than the remainder of the genome.

Unexpectedly, we find RNA pol II levels and nucleosome occupancy to correlate not only at TSRs but also upstream of most genes, even within PTUs, pointing to a role for nucleosome positioning in RNA processing, rather than transcription initiation.

## Results

### RNA pol II transcription initiates transcription at the 5′-end of H2A.Z peaks

In *T. brucei*, RNA pol II transcription is thought to initiate within ~10-kb-wide regions enriched in H2A.Z (Siegel *et al*, 2009; Kolev *et al*, 2010). However, in an effort to identify RNA pol II promoter elements, we decided to determine the sites of RNA pol II transcription initiation more precisely following a two-pronged approach. First, we determined the genome-wide distribution of RNA pol II enrichment by ChIP-seq (chromatin immunoprecipitation followed by next-generation sequencing) using a cell line expressing

Ty1-tagged *RPB9*, a subunit only found in the RNA pol II complex (Devaux *et al*, 2006). In addition, we determined the distribution of H2A.Z by MNase-ChIP-seq, using a custom-made, H2A.Z-specific antibody (Appendix Fig S1). The RPB9-ChIP-seq data revealed RNA pol II to be strongly enriched across a ~2-kb region at the 5′-end of H2A.Z-enriched sites (Fig 1A). Since polymerase enrichment correlates with transcriptional pausing and RNA pol II pausing just downstream of promoters has been described in a wide range of organisms (Adelman & Lis, 2012), we hypothesized that the sites of RNA pol II enrichment observed in this study mark sites of transcription initiation.

Next, to corroborate that sites of RNA pol II enrichment indeed mark sites of transcription initiation, we mapped the distribution of small primary transcripts carrying a 5′-triphosphate. Since triphosphates are removed during RNA maturation, mapping of triphosphate-containing RNA can be used to identify sites of transcription

initiation (Fig 1B). Selecting for short triphosphate-containing transcripts allowed us to reduce the amount of rRNA and to identify sites of transcription initiation more precisely. As anticipated, we found small primary transcripts to be enriched at the 5′-end of sites of H2A.Z enrichment (Fig 1C). Interestingly, levels of primary transcripts peaked just upstream of RNA pol II enrichment (compare Fig 1A and C), suggesting that RNA pol II transcription pauses 100–200 bp downstream of its initiation, similar to the situation observed in metazoans (Adelman & Lis, 2012). However, unlike dispersed promoters in metazoans, which induce bidirectional transcription initiation (Wu & Sharp, 2013), our data indicate that in *T. brucei*, transcription initiation has a strong strand bias with a ratio of sense to antisense primary transcripts of 4:1.

Together, our RNA pol II ChIP-seq data and our primary transcripts analysis revealed the 5′-end of H2A.Z-enriched regions as the primary sites of RNA pol II initiation.

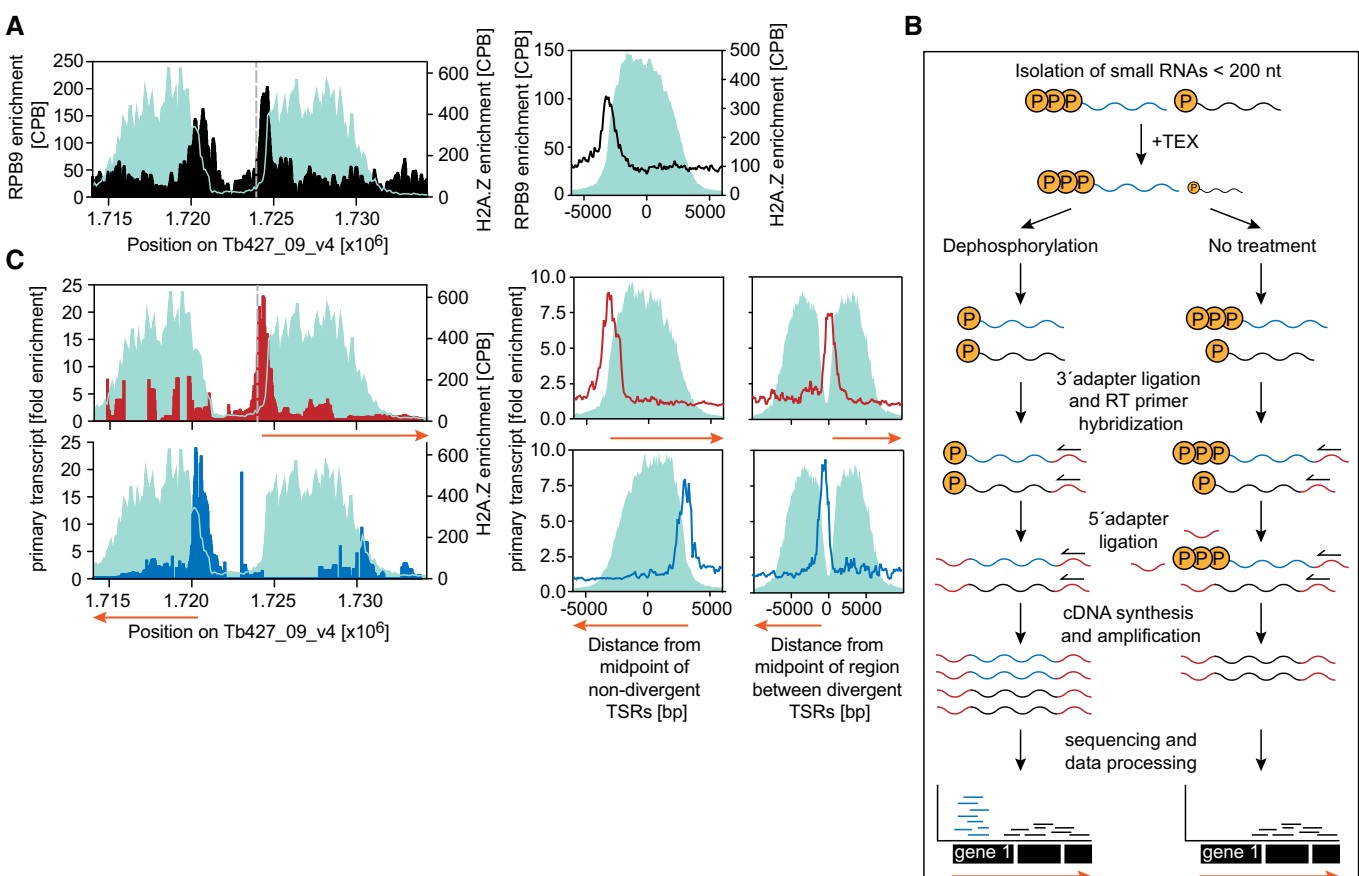

**Figure 1.  Identification of transcription initiation sites in *Trypanosoma brucei*.**

A   ChIP-seq data of the RNA pol II subunit RPB9 are shown across a representative divergent TSR of chr. 9 (left panel) and averaged across all non-divergent TSRs (*n* = 57) in black (right panel). To illustrate the width of the TSR, H2A.Z enrichment determined by MNase-ChIP-seq is shown in cyan.

B   Outline of small 5′-triphosphate-RNA-seq. Small total RNA < 200 nt was purified from *T. brucei* and enriched for primary transcripts containing a 5′-triphosphate (blue RNA) by depletion of 5′-monophosphate-containing RNA (black RNA) using a Terminator 5′-Phosphate-Dependent Exonuclease (TEX). To identify undigested monophosphate-containing RNA contaminants, the sample was split and libraries were prepared from 5′-polyphosphatase-treated and untreated material.

C   Primary transcripts derived from the top strand are shown in red across a representative divergent TSR of chr. 9 (top left panel), averaged across 27 non-divergent TSRs (top middle panel) and averaged across 71 divergent TSRs (top right). Primary transcripts derived from the bottom strand are shown in blue across the same TSR (bottom left panel), averaged across 30 non-divergent TSRs (bottom middle panel) and averaged across 71 divergent TSRs (bottom right). The gray dashed line is added to illustrate the shift in RNA pol II and primary transcript enrichment. Orange arrows indicate the direction of transcription. H2A.Z enrichment is shown in cyan.

                                                                           

### The ability of DNA sequence elements to initiate transcription depends on the genomic context

To test whether a specific DNA sequence element can drive transcription initiation *in vivo* in *T. brucei*, we devised a reporter assay that allowed us to insert different DNA sequence elements upstream of a firefly luciferase reporter gene (*FLUC*). This required the identification of a non-transcribed, yet transcriptionally permissive region of the genome. The organization of the *T. brucei* genome into long PTUs means that most of the core genome is actively transcribed, the exceptions being small regions between PTUs (Kolev *et al*, 2010; Siegel *et al*, 2010). However, previous ChIP-seq assays had indicated that polycistronic transcription terminates in regions enriched in H3.V and H4.V (Fig 2A), leading to the assumption that these histone variants repress transcription (Siegel *et al*, 2009). Therefore, we inserted the reporter construct between two divergent PTUs,

which corresponds to a non-transcribed region of the genome that contains low levels of H3.V and H4.V (Fig 2B).

To determine the capacity of DNA sequence elements to initiate transcription, we generated two reporter constructs each containing the DNA from a different transcription start region (TSR) upstream of a luciferase gene (regA and regB; Fig 2C, Table EV1). Insertion of the reporter constructs into a region between two divergent PTUs of chr. 1 resulted in an 8.7-fold and 9.6-fold increase in luciferase activity compared to insertion of a control reporter construct lacking a promoter sequence. These results indicated that the DNA sequence found at TSRs contained elements able to initiate transcription. Targeting these luciferase reporter constructs to two genomic loci with increased H3.V and H4.V levels did not result in luciferase expression, not even in a cell line lacking H3.V (Fig EV1), underlining the importance of the genomic context for gene expression.

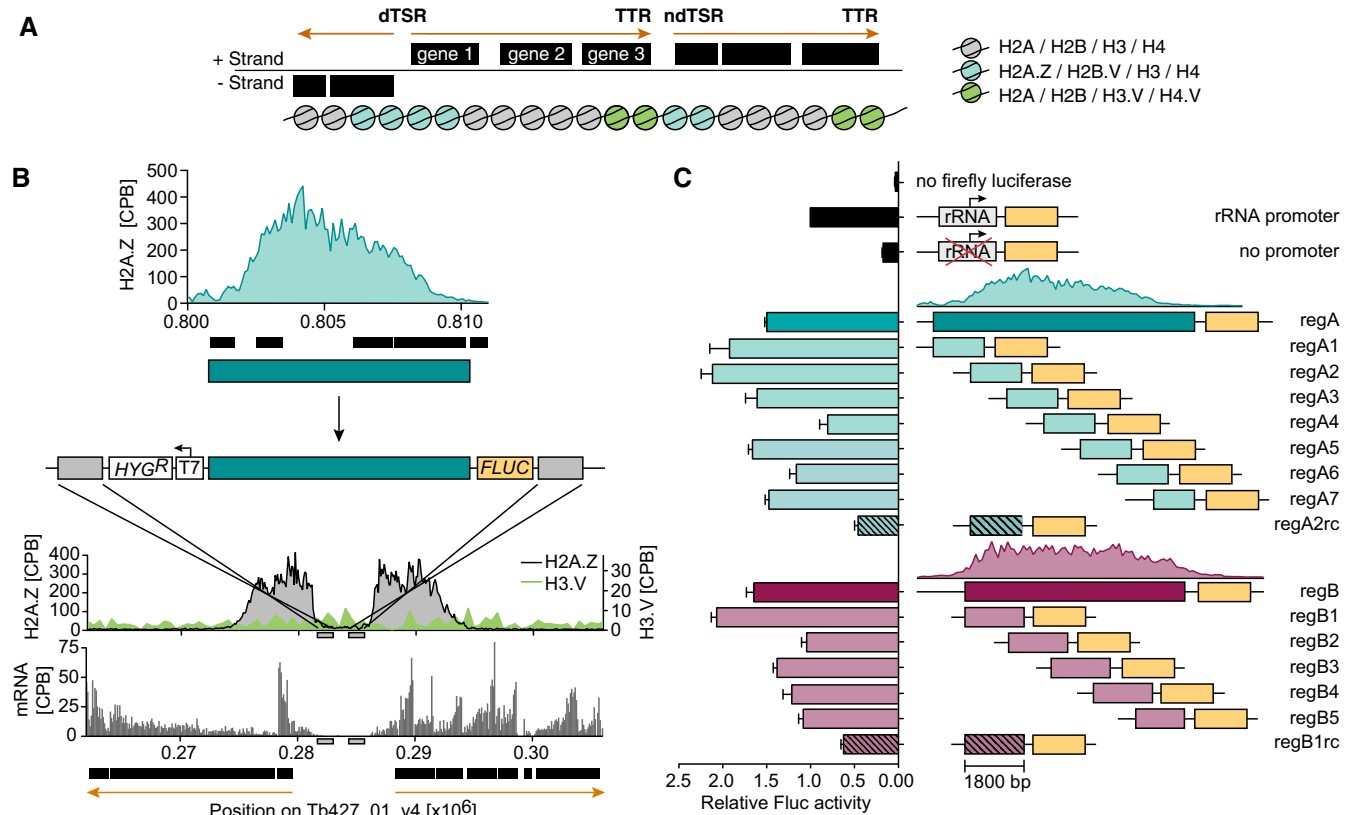

**Figure 2.** *In vivo* reporter assay reveals promoter activity of distinct sequence elements.

A  Outline of the genome organization. Boundaries of PTUs are marked by nucleosomes containing different types of histone variants. H2A.Z and H2B.V (cyan nucleosomes) are located at divergent (dTSR) and non-divergent transcription start regions (ndTSR). H3.V and H4.V (green nucleosomes) are located at transcription termination regions (TTRs). Orange arrows indicate the direction of transcription.

B  Outline of reporter assay. A region enriched in H2A.Z (H2A.Z MNase-ChIP-seq data are shown as counts per billion reads, CPB; cyan), which we defined as TSR, was cloned upstream of a firefly luciferase gene (*FLUC*). The reporter construct was targeted to a non-transcribed locus between a dTSR of chr. 1 (mRNA levels are shown in gray and were determined previously, Vasquez *et al*, 2014), containing low levels of H3.V (H3.V levels are shown in green and were determined previously, Siegel *et al*, 2009). The luciferase gene cassette consists, from 5′ to 3′, of *trans*-splicing motifs and 5′ UTR from a GPEET gene, the luciferase CDS, and the 3′ UTR of aldolase including a polyadenylation site. Gray boxes represent regions of homology.

C  Luciferase assays were performed after insertion of complete or partial TSR DNA sequences. Two different TSR DNA sequences were analyzed (regA, blue; regB, pink). Striped bars represent the respective fragment inserted as reverse complement (regA2rc, regB1rc). To account for differences in cell number, Fluc activity was normalized to ectopically expressed *Renilla* luciferase activity. To account for technical variations, values were normalized to rRNA promoter-driven Fluc activity. Data are presented as mean ± SD. Error bars indicate standard deviation between two replicates.

To identify specific DNA elements that are able to initiate transcription within the TSR, we divided the two TSRs into seven and five evenly spaced fragments, respectively (1,800 bp in width and with 500-bp overlap between adjacent fragments). When repeating the assay using the different TSR fragments, we observed that all 12 fragments were capable of initiating transcription well above background (Fig 2C). The highest luciferase levels were observed for the two fragments originating from the 5′-end of each TSR (regA2 and regB1). One fragment extending ~300 nt upstream of the TSR (regA1) yielded similar levels.

The observation that all TSR fragments were capable of initiating transcription at similar levels argues against the presence of well-defined canonical promoter motifs. Instead, the observed pattern is similar to that reported for dispersed promoters that lead to broad regions of transcription initiation (Deaton & Bird, 2011).

## GT-rich promoter elements can induce transcription

Our analysis of primary transcripts indicated that transcription initiation has a strong strand bias. To determine whether TSR-derived sequence elements are sufficient to ensure directionality (i.e., more sense transcription than antisense transcription), we inverted TSR fragments in the reporter assays (Fig 2C; regA2rc, regB1rc). After inversion, both fragments showed a 4.7-fold and 3.3-fold decrease in their capacity to drive luciferase expression, in good agreement

with the 4:1 ratio of sense to antisense primary transcripts we observed. Based on these results, we hypothesized the following: (i) Promoter elements capable of ensuring directional transcription initiation are present in *T. brucei*, (ii) these elements are enriched within TSRs, and (iii) they are unevenly distributed across the coding and non-coding strands, giving rise to the observed bias in sense vs. antisense transcripts.

To identify such sequence elements, we divided all TSRs into five evenly spaced regions, and in each region, we searched for 10mers that were unevenly distributed between the two strands, that is, 10mers that were present at least sixfold more often on the coding than on the non-coding strand. The 5′-end of TSRs contained the highest number of unevenly distributed 10mers, which decreased toward the 3′-end of the TSR (Fig 3A and Dataset EV1). The vast majority of enriched 10mers contained long stretches of Gs or Ts (Fig 3B and Dataset EV1), which is in good agreement with previous reports of a G-to-C and T-to-A skew in *L. major* and *T. brucei* (McDonagh *et al*, 2000; Siegel *et al*, 2009).

To test our hypothesis that these GT-rich elements are capable of driving directional transcription, we generated two synthetic promoters (210 and 416 bp) composed of 10mers that are enriched on the coding strand (Appendix Fig S2, Table EV2). Integration of the synthetic GT-rich promoters upstream of the *FLUC* gene yielded high luciferase activity, 2.4-fold and 2.0-fold higher than full-length TSRs (compare Figs 2C and 3C). Furthermore, transcription was

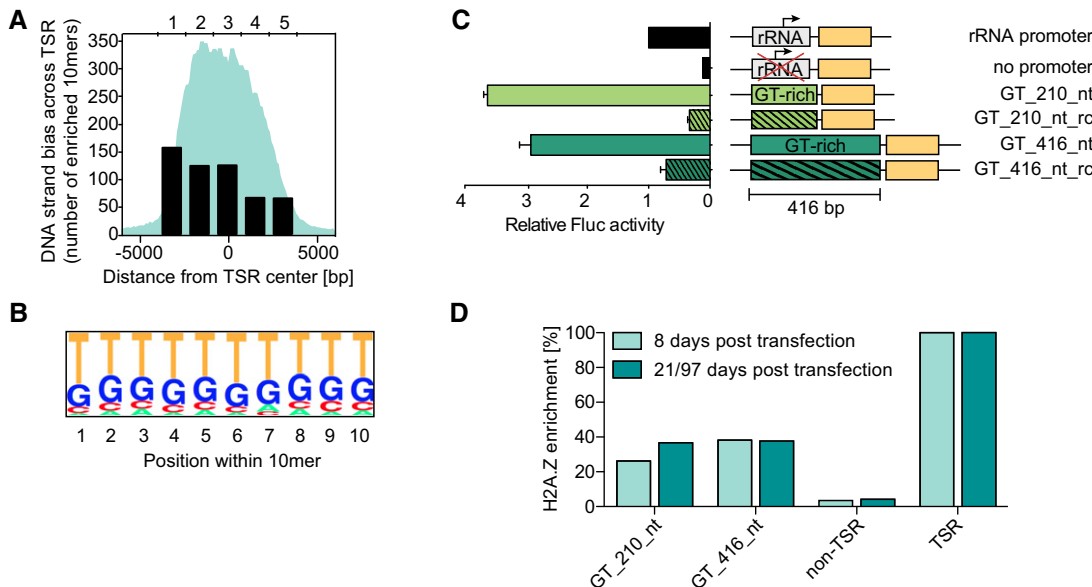

**Figure 3.  GT-rich sequences located on the coding strand of TSRs mediate transcription.**

A  To identify strand biases, each TSR sequence was divided into five regions of equal size and the number of 10mers enriched at least sixfold on the coding strand compared to the non-coding strand was counted. H2A.Z enrichment is illustrated in cyan.

B  The consensus sequence of the 10mers enriched on the coding strand within region 1 was calculated using pictogram (Burge *et al*, 1999).

C  Firefly luciferase assays were performed after insertion of two synthetic GT-rich sequences (GT_210_nt in light green and GT_416_nt in dark green) composed of the most enriched 10mers (Appendix Fig S2). Striped bars represent the reverse complement sequences (GT_210_nt_rc and GT_416_nt_rc). To account for differences in cell number, Fluc activity was normalized to ectopically expressed *Renilla* luciferase activity. To account for technical variations, values were normalized to rRNA promoter-driven Fluc activity. Data are presented as mean ± SD. Error bars indicate standard deviation between two replicates.

D  Relative H2A.Z levels (based on MNase-ChIP-seq) across GT_210_nt, GT_416_nt, a 6-kb region upstream of the TSR (non-TSR), and the TSR upstream of the site of insertion. MNase-ChIP-seq of GT_210_nt cells and GT_416_nt cells was performed 8 and 21 days post-transfection (dpt) and 8 and 97 dpt, respectively. The H2A.Z level of the adjacent TSR was set to 100%.

highly dependent on the direction of the GT-rich element, supporting our hypothesis that DNA sequence elements confer directional transcription. Insertion of GT-rich sequence elements between divergent TSRs had no effect on the transcript levels of genes upstream or downstream of the insertion site (Fig EV2).

To compare the activity of the GT-rich sequence elements to that of endogenous RNA pol I and RNA pol II transcription, we targeted an rRNA promoter-driven luciferase gene to the rDNA spacer and inserted a promoter-less luciferase gene within a RNA pol II-transcribed PTU. Not surprisingly, we find rRNA promoter-driven luciferase expression to be 38.7- to 193.6-fold higher when the transgene is located in the rDNA spacer compared to the region between two divergent RNA pol II TSRs. In agreement with earlier studies (Alsford *et al*, 2005), we find luciferase levels to vary greatly among clones, probably because the constructs integrated in different genomic locations. The large differences in transgene expression between insertion in the rDNA spacer and between divergent TSRs can be explained by the spatial restriction of RNA pol I to the nucleolus.

For RNA pol II-driven luciferase, we find the endogenous levels to be 11.6-fold higher than those induced by the GT-rich element and the GT-rich element-induced levels 18.5-fold higher than those from our negative control, not containing any promoter upstream of the luciferase gene (Fig EV3).

Thus, our results indicate that GT-rich elements can ensure directional transcription initiation, albeit not at the levels of endogenous RNA pol II transcription, which likely reflects the influence of the genomic location.

### GT-rich promoter elements induce targeted deposition of H2A.Z

Previous studies have reported that RNA pol II transcription initiation in *T. brucei* occurs in regions that lack promoter elements, suggesting that a permissive chromatin structure, which allows RNA pol II to access the DNA, may be sufficient for transcription initiation (Marchetti *et al*, 1998; McAndrew *et al*, 1998). In support of this hypothesis, we have previously found TSRs to be enriched in specific different histone variants and histone modifications, including the histone variant H2A.Z (Siegel *et al*, 2009; Wright *et al*, 2010).

To determine whether GT-rich promoter elements contribute to the enrichment of histone variants at TSRs, we mapped the genome-wide distribution of H2A.Z in transgenic cell lines containing GT-rich sequence elements of two different lengths. The data revealed that both sites containing synthetic GT-elements were enriched in H2A.Z. However, levels of H2A.Z were lower across GT-elements than across an endogenous TSR (Fig 3D).

Next, we investigated the ability of GT-rich elements to drive transcription and to promote H2A.Z deposition over time. Chromatin structures are dynamic, and it is possible that following insertion of the GT-rich element, it may take several cell divisions for transcription levels and H2A.Z levels to peak. Therefore, we re-generated the cell lines containing the short and the long GT-rich promoter construct, repeated the luciferase assays 8 and 30 days post-infection, and observed a significant change in luciferase expression over time (Fig EV4). In addition, we repeated the H2A.Z MNase-ChIP-seq assay 8 days post-infection. For the short GT-rich element, we saw a time-dependent increase in H2A.Z levels, corresponding to the increase in luciferase activity. For the long GT-rich element, no such increase was observed (Fig 3D).

Thus, while GT-rich sequence elements can induce transcription and promote the targeted deposition of H2A.Z, the degree to which it does may vary over time. In addition, our data indicate that relocated TSR sequences and GT-rich promoter-induced transcription and/or H2A.Z levels do not reach those of endogenous RNA pol II promoters, highlighting again the importance of the genomic context.

### Sites enriched in H2A.Z show increased sensitivity to MNase

Numerous studies performed in different organisms, including *T. brucei,* have suggested that nucleosomes containing H2A.Z are less stable than canonical nucleosomes (Suto *et al*, 2000; Abbott *et al*, 2001; Zhang *et al*, 2005; Jin & Felsenfeld, 2007; Siegel *et al*, 2009). To understand the biological significance of decreased nucleosome stability, we determined whether DNA associated with H2A.Z, that is, DNA within TSRs, is more accessible to proteins than DNA outside of TSRs.

To assess DNA accessibility, we established an MNase-ChIP-seq approach for *T. brucei*. This approach combines MNase digestion of formaldehyde-cross-linked chromatin with immunoprecipitation of nucleosomes and high-throughput paired-end DNA sequencing (Fig EV5A). MNase preferentially cleaves within linker DNA between nucleosomes. The resulting MNase digestion products, 147 bp in length, accurately reveal the positions of nucleosomes when they are sequenced and mapped back to the genome and can be used to generate nucleosome occupancy maps (Cole *et al*, 2012). In addition, MNase-ChIP-seq can be used to identify "loosely bound" nucleosomes. Regions containing loosely bound nucleosomes and, as a consequence, more accessible DNA yield on average shorter MNase cleavage products (< 147 bp) than regions that are composed of more compact chromatin (Weiner *et al*, 2010).

To avoid over-digestion, we titrated MNase digestion such that a small population of dinucleosomal DNA remained (Fig EV5B and C). Next, nucleosomes were immunoprecipitated with a custom-made histone H3 antiserum (Gassen *et al*, 2012), and the DNA was isolated and sequenced. A total of 18.6 million concordantly aligning 100-bp sequence reads could be mapped to the *T. brucei* genome corresponding to an average genome coverage of ~53×.

Despite *T. brucei* utilizing "unusual" mechanisms of gene regulation, the biophysical properties of DNA should be conserved. Given that dinucleotides vary considerably with respect to their bending properties, optimal nucleosome formation occurs when bendable dinucleotides (AT and TA) occur at intervals of 10 bp. The exact positioning of the histone octamer with respect to the ~10-bp helical repeat is termed "rotational positioning" (Struhl & Segal, 2013). Thus, to validate the quality of our nucleosome occupancy maps, we determined the rotational positioning of *T. brucei* nucleosome sequences. As described for all organisms analyzed so far, we found a strong periodicity in AA/AT/TT/TA (Fig EV5D). These data indicate that dinucleotide patterns are important for the rotational positioning of trypanosome nucleosomes and suggest that our nucleosome occupancy maps are of high resolution.

Analysis of nucleosome occupancy across TSRs indicated no general depletion of nucleosomes. However, grouping of sequenced MNase digestion products based on size revealed that TSRs are

enriched in short sub-mononucleosomal fragments (100–130 nt) and depleted from long supra-mononucleosomal fragments (> 175 nt) (Fig 4A). To validate this observation, we plotted the averaged nucleosome occupancy across all TSRs. This further supported an increase in sub-mononucleosomal DNA fragments across TSRs (Fig 4B).

These findings indicate that DNA associated with H2A.Z-containing nucleosomes is more easily digested than DNA bound to canonical nucleosomes, revealing TSRs as regions of increased DNA accessibility.

## Nucleosome occupancy at exon boundaries correlates with RNA pol II enrichment and transcript levels

From humans to yeast, promoters that drive the transcription of ubiquitously expressed genes contain a well-defined nucleosome-depleted region (NDR) flanked by strongly positioned nucleosomes at the site of transcription initiation (Hughes & Rando, 2014). In addition to TSSs, strongly positioned nucleosomes have been observed at intron/exon boundaries in many different eukaryotes including other protozoan parasites such as *Plasmodium falciparum* (Schwartz *et al*, 2009; Tilgner *et al*, 2009; Kensche *et al*, 2016). Generally, exons show increased nucleosome occupancy compared to introns, and it has been proposed that this increase temporarily slows the rate of RNA pol II transcription, thereby facilitating co-transcriptional recruitment of splicing factors (Naftelberg *et al*, 2015).

Given the broad regions of ubiquitous transcription initiation, we investigated whether *T. brucei* may contain a well-defined NDR

within each TSR. In the absence of sharp peaks of transcription initiation and because genes are organized in PTUs, we averaged the nucleosome occupancy across TSRs using the ATG of the first gene of each PTU as a proxy for TSSs. This analysis revealed a strong depletion of H3 and H2A.Z-containing nucleosomes ~90 nt upstream of the ATG (Fig 5A and B). However, our primary transcript data had suggested that RNA pol II transcription initiation occurs at the 5′-end of H2A.Z-enriched regions, which is much further upstream than the ATG of the first gene. In addition, given that the median 5′ UTR length in *T. brucei* is ~90 nt (excluding the spliced-leader RNA) (Siegel *et al*, 2011), the center of the NDR coincided with the 5′-end of the 5′ UTR. Thus, our data point to a role of NDRs in RNA processing rather than transcription initiation.

With the exceptions of two genes, no *cis*-splicing has been observed in *T. brucei* (Mair *et al*, 2000; Berriman *et al*, 2005). Nevertheless, during *trans*-splicing an identical 39-nt spliced-leader sequence from the donor-spliced-leader RNA is transferred to the 5′-end of every mRNA (Michaeli, 2011) and, the 5′-end of the 5′ UTR of each gene serves as a splice acceptor site (SAS). Thus, should strongly positioned nucleosomes mark the exon boundary, rather than sites of transcription initiation, one would expect to find an NDR upstream of every gene, not just in TSRs. We therefore extended the analysis of nucleosomes to the remaining genes of PTUs (all genes except the first gene) and observed a similar pattern of positioning, indicating that most genes are preceded by an NDR (Fig 5C).

Should nucleosomes be able to slow the rate of RNA pol II elongation and thereby enhance splicing efficiency as previously proposed (Naftelberg *et al*, 2015), we would expect (i) RNA pol II

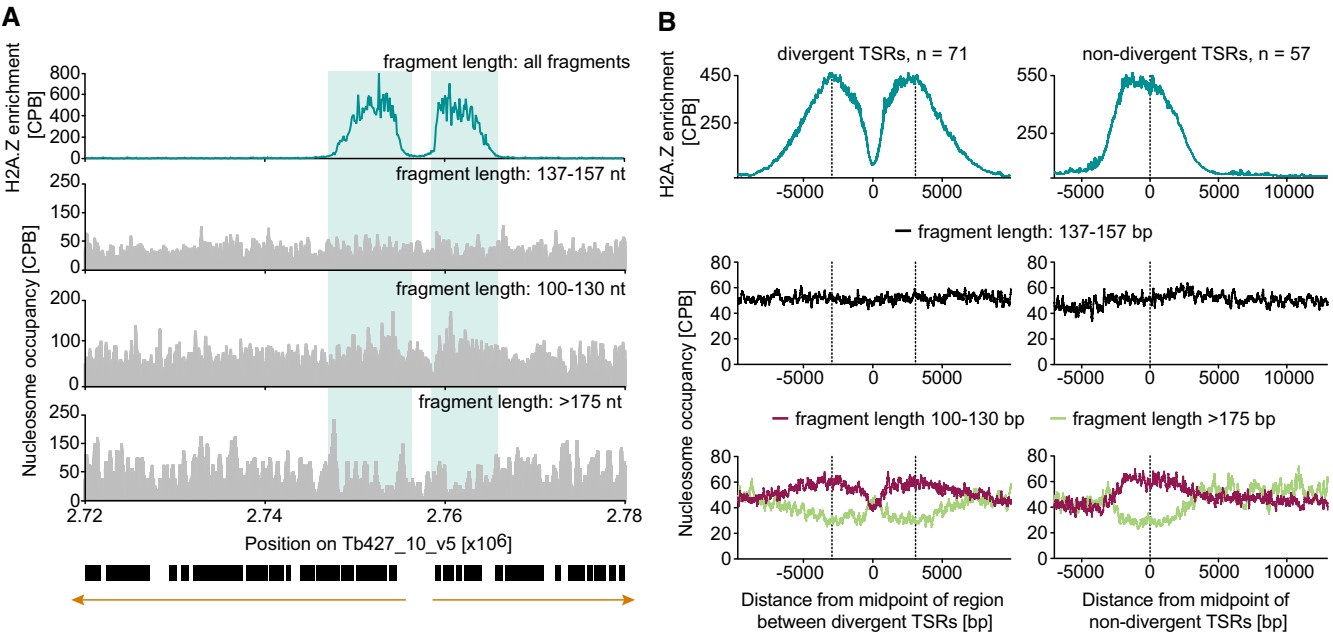

**Figure 4.  TSRs exhibit increased MNase sensitivity.**

A   MNase-ChIP-seq data of H2A.Z-containing mononucleosomes and total mononucleosomes (nucleosome occupancy) grouped based on size of digestion products (for outline, see Fig EV5). Black boxes represent open reading frames. Orange arrows indicate the direction of transcription. Shown is a representative TSR of chr. 10.

B   The enrichment of H2A.Z and total nucleosome occupancy averaged across all divergent TSRs (left panel) and non-divergent TSRs (right panel) are plotted relative to the midpoint of the region between the TSRs and the TSR center, respectively. Dashed lines mark the respective TSR centers.

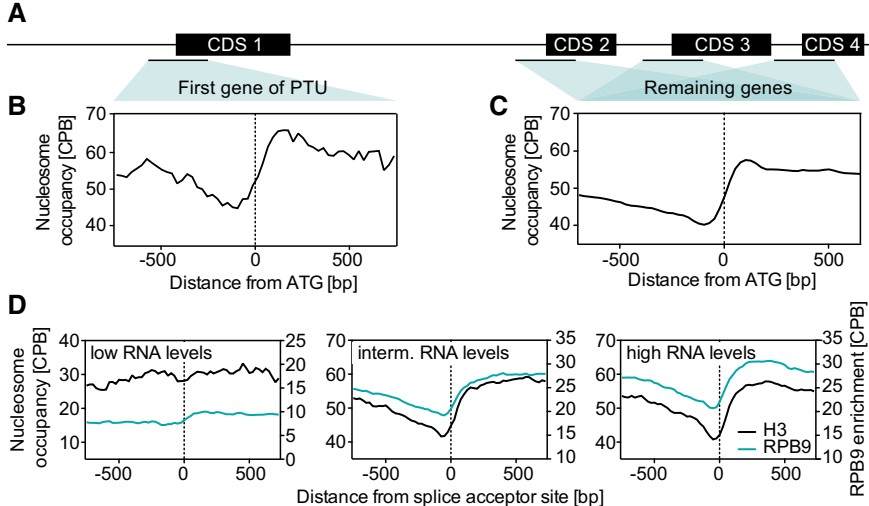

**Figure 5.  Nucleosome depletion correlates with the level of gene expression.**

A   Schematic display of a PTU.

B   Nucleosome occupancy is plotted relative to the start codon (ATG) of the first gene of a PTU and averaged across all PTUs (*n* = 184). The definition of the first gene of a PTU is based on a previous study (Kolev *et al*, 2010) and genome version Tb927v24.

C   Total nucleosome occupancy is plotted relative to the ATG and averaged across all genes except the first gene of a PTU (*n* = 12,220).

D   Nucleosome occupancy is plotted relative to the splice acceptor sites and averaged across the 25% of genes containing the highest RNA levels (left panel, *n* = 690), the 25% of genes containing intermediate RNA levels (middle panel, *n* = 690), and the 25% of genes containing the lowest RNA levels (lower panel, *n* = 690). RNA levels were determined previously (Fadda *et al*, 2014).

levels to correlate with nucleosome occupancy and (ii) a strong increase in nucleosome occupancy to correlate with efficient *trans*-splicing and, as a consequence, high RNA levels. Analyzing our RNA pol II ChIP-seq data, we find RNA pol II levels to closely resemble those of nucleosome occupancy (Fig 5D). In addition, when grouping genes based on the level of detected transcripts, high (top 25%), intermediate (middle 25%), and low (bottom 25%), we find a well-defined NDR upstream of genes with high and intermediate transcript levels, but not upstream of genes with low levels (Fig 5D).

Taken together, our data suggest that while TSRs lack well-defined NDRs in *T. brucei*, strong nucleosome positioning across exon boundaries may affect the rate of RNA pol II and RNA processing.

## Composition of polyY tract affects nucleosome positioning and gene expression

One important contributor to splicing is the pyrimidine-rich (polyY) tract, located just upstream of the SAS. At the RNA level, polyY tracts serve as binding sites for the U2AF65 subunit of the spliceo-some (Kielkopf *et al*, 2001). In addition, at the DNA level, homopolymeric sequences such as polyY tracts are intrinsically rigid and are thus strongly inhibitory to nucleosome formation (Suter *et al*, 2000).

Previous transient transfection experiments, in which the effect of nucleosome positioning could not be evaluated, indicated that length and composition of polyY tracts influence *trans*-splicing in *T. brucei* (Siegel *et al*, 2005). To better understand the cause of nucleosome positioning at exon boundaries, we investigated the influence of the polyY tract in nucleosome positioning *in vivo*. To

this end, we generated three transgenic cell lines carrying a luci-ferase reporter construct containing the short GT-rich promoter followed by (i) a polyY tract found upstream of the highly expressed GPEET procyclin genes, (ii) a long T-rich polyY tract shown to be among the most efficient in transient *trans*-splicing assays, or (iii) no polyY tract. All constructs contained an identical GPEET 5′ UTR sequence. Luciferase levels were highest for the GPEET polyY tract, twofold lower for the T-rich polyY tract, and virtually absent in the cell line lacking a polyY tract (Fig 6A). These measurements are in good agreement with transient trans-fection experiments and underline the importance of polyY tracts for efficient *trans*-splicing.

Next, we used these cell lines to generate three additional high-resolution nucleosome occupancy maps (as described above). The additional nucleosome occupancy maps were interesting in several aspects. First, we noticed that in all three cell lines, the GT-rich sequence was strongly depleted of nucleosomes (Fig 6B). We suspect this depletion to be caused by the long homopolymeric G and T stretches present in the promoter element. The low overall amount of nucleosomes in this region may also explain why the H2A.Z levels at GT-rich regions did not reach those of endogenous TSRs (Fig 3D).

Comparing the three additional nucleosome occupancy maps, it became apparent that the homopolymeric nature of the long T-rich polyY tract, which should give rise to a rather rigid DNA double helix, resulted in an expansion of the NDR (Fig 6B, middle panel) compared to the nucleosome occupancy observed in the cell line carrying the GPEET polyY tract (Fig 6B, upper panel) and that carry-ing no polyY tract (Fig 6B, lower panel). Finally, our data revealed the presence of an NDR upstream of the ORF in the absence of a polyY tract (Fig 6B, lower panel).

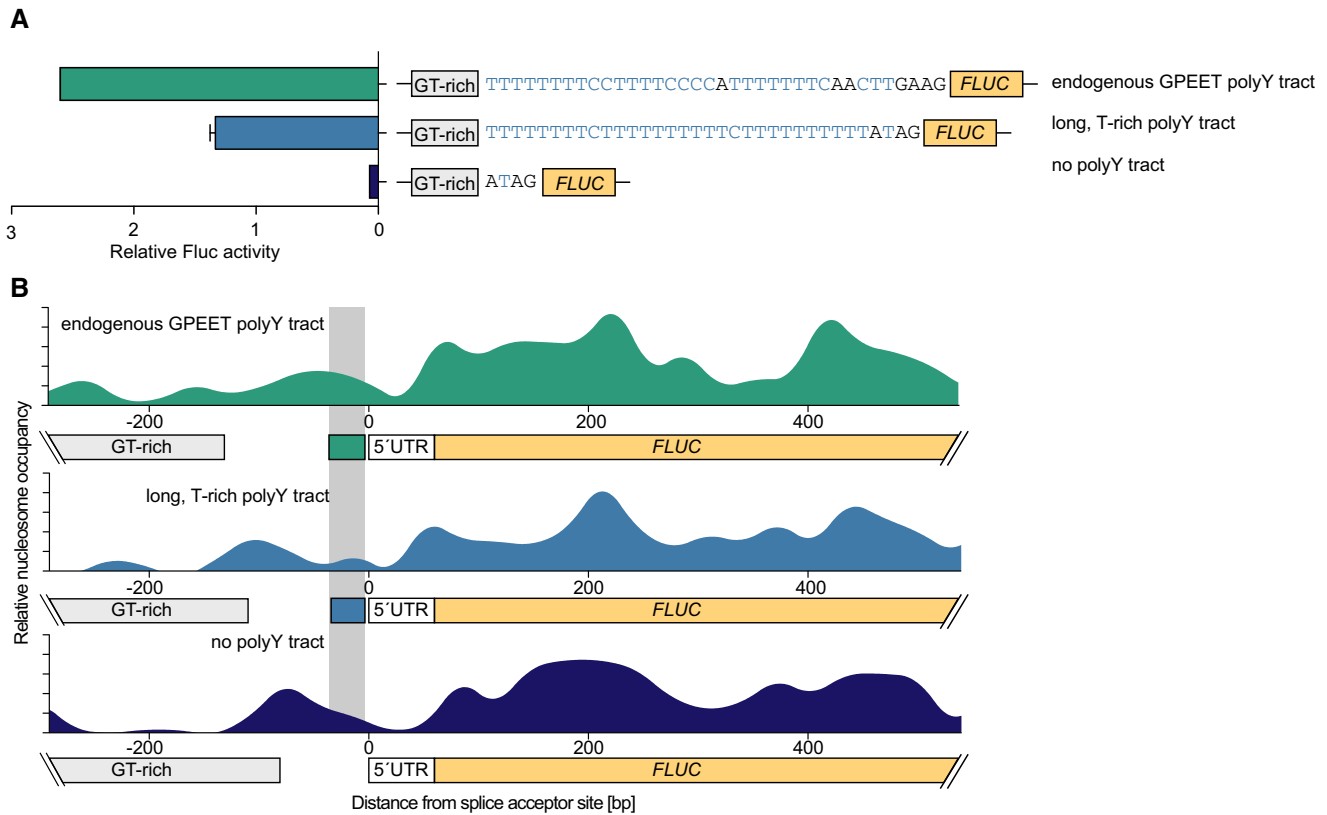

**Figure 6.  Composition of polyY tract affects gene expression and nucleosome positioning.**

A   Luciferase assays were performed after insertion of a *FLUC* reporter construct containing the short GT-rich promoter and either the endogenous GPEET polyY tract (green), a long T-rich polyY tract (light blue), or no polyY tract (dark blue). To account for technical variations, values were normalized to rRNA promoter-driven Fluc activity. Data are presented as mean ± SD. Error bars indicate standard deviation between two replicates.

B   Nucleosome occupancy was determined for the three cell lines described in (A). The maps are aligned to the respective splice acceptor site (position 0). The maps were generated from histone H3 MNase-ChIP-seq data processed with bowtie 1.1.1 (Dataset EV2) and default nucwave settings (Quintales *et al*, 2015). The location of the endogenous GPEET polyY tract is highlighted in gray.

Thus, while the polyY tract is clearly important for efficient *trans*-splicing and can affect nucleosome positioning, there must be other elements that ensure the presence of NDRs. Such elements may be located in the 5′ UTR.

## Discussion

Several recent genome-wide studies performed in different higher eukaryotes revealed a strong association of ubiquitously expressed genes with dispersed promoters, lacking well-defined sequence motifs. To elucidate the conservation of this association, we analyzed promoter motifs in *T. brucei,* an evolutionarily highly divergent eukaryote that lacks transcriptional regulation of RNA pol II-transcribed genes. In addition, promoters and associated transcription factors have remained elusive in *T. brucei*; hence, how RNA pol II transcription is initiated in this parasite remains a fundamental question.

In this study, we have identified a GT-rich promoter element capable of conferring transcription initiation and contributing to the targeted deposition of H2A.Z. Our data demonstrate that RNA pol II promoter sequence elements are present even in organisms lacking regulation of RNA pol II transcription initiation. Thus, our findings suggest that the link between a lack of well-defined sequence elements and a lack of gene regulation may be highly conserved in evolution.

In contrast to the general conservation we have found at the DNA level, our data revealed significant differences regarding the chromatin structure at promoter sites. Unlike NDRs in other eukaryotes, in *T. brucei* we have shown NDRs to be associated with splice sites and not with sites of transcription initiation. Given the regulatory potential of NDRs, this difference may reflect the lack of transcriptional regulation in *T. brucei* and the importance of post-transcriptional mechanisms of gene regulation, such as *trans*-splicing. The observed link between nucleosome occupancy, polymerase levels, and splicing elements is not unprecedented. Genome-wide nucleosome occupancy maps not only revealed NDRs across promoters, but also exposed a general increase of nucleosome occupancy in exons compared to introns and a preference of nucleosomes for constitutively spliced exons over alternatively spliced exons (Schwartz *et al*, 2009; Huang *et al*, 2012). Splicing of exons occurs co-transcriptionally and so it has been proposed that

                                                            

nucleosomes positioned at the 5′-end of exons may function as "speed bumps" to slow down RNA pol II elongation, thereby promoting inclusion of exons (Schwartz & Ast, 2010). According to this model, the presence or absence of nucleosomes at intron/exon boundaries influences exon selection, thereby affecting alternative splicing.

*Cis*-splicing and *trans*-splicing share many mechanistic similarities (Michaeli, 2011). Yet, mapping of SASs in *T. brucei* revealed an unexpected heterogeneity in SASs. For many genes, the major SAS was found to differ from one life cycle stage of the parasite to another, leading to the generation of transcripts with different 5′ UTRs (Kolev *et al*, 2010; Nilsson *et al*, 2010; Siegel *et al*, 2010). Thus, in the absence of transcriptional regulation, alternative *trans*-splicing has been proposed as a mechanism to regulate gene expression. The factors contributing to life cycle-specific SAS preferences are still unknown.

As reported for organisms that undergo *cis*-splicing, our data reveal a strong increase in nucleosome occupancy at the 5′-end of exons. In addition, our RNA pol II ChIP-seq data suggest that the increase in nucleosome occupancy may act as a barrier and temporarily slow RNA pol II elongation leading to the observed increase in RNA pol II levels. A decrease in RNA pol II elongation speed may increase *trans*-splicing efficiency. Furthermore, small changes in nucleosome positioning may slow the polymerase at different positions, thereby affecting the choice of SAS. Analogously, small differences in nucleosome positioning between different life cycle stages could contribute to the observed life cycle-specific SAS preferences. These hypotheses could be corroborated in the future by comparing nucleosome occupancy maps from different life cycle stages of *T. brucei*.

One critical finding of our promoter assays was the importance of the genomic context; that is, we found the ability of specific DNA sequence elements to initiate transcription to vary greatly depending on the site of insertion. For example, insertion of an rRNA promoter-driven luciferase gene between two divergent TSRs yielded low levels of luciferase, while insertion of the same construct into rDNA arrays yielded high levels. Insertion of a TSR sequence between two divergent TSRs led to intermediate levels of transcriptional activity, while insertion of the same construct into a region enriched in H3.V led to no activity. Insertions of GT-rich elements yielded similar site-specific effects.

A reason for the importance of the genomic context could be that the *T. brucei* genome possesses a well-defined 3D genome architecture. This hypothesis is supported by several microscopy-based studies that have observed clustering of specific RNA pol I, RNA pol II, and RNA pol III subunits in distinct loci (Navarro & Gull, 2001; Uzureau *et al*, 2008; Alsford & Horn, 2011). Most notably, the distribution of RNA pol I is restricted to the nucleolus and the expression site body where the variant surface antigens are transcribed (Navarro & Gull, 2001). Thus, the target site-specific differences in RNA pol I-driven luciferase activity are in good agreement with an important role of genome organization.

The lack of transcriptional regulation, the organization of genes in long PTUs, and the apparent absence of promoter motifs in trypanosomatids have suggested fundamental differences in terms of gene regulation compared to other eukaryotes. In this study, we find that specific DNA sequence elements can drive directional transcription and affect local chromatin structure. Furthermore, our data

underscore the importance of genomic context in gene expression and establish a link between nucleosome positioning and exon boundaries.

Thus, our findings suggest that despite its evolutionary divergence, *T. brucei* uses several of the mechanisms found in other eukaryotes to regulate its gene expression, implying that many of the strategies to regulate gene expression are highly conserved in evolution. At the same time, our data revealed fundamental differences regarding the patterns of nucleosome occupancy between higher eukaryotes and trypanosomes. The observed differences may reflect the lack of RNA pol II transcription regulation in *T. brucei* and its strong dependency on post-transcriptional mechanisms of gene regulation.

## Materials and Methods

For details of *T. brucei* manipulation, plasmids, and data analysis, see the Appendix Supplementary Methods.

### *Trypanosoma brucei* culture

*Trypanosoma brucei* wild-type and genetically modified strains were derived from Lister 427 bloodstream-form MITat 1.2 (clone 221) or a derivative "single marker" cell line (SM) expressing T7 RNA polymerase and the Tet repressor (Wirtz & Clayton, 1995) and cultured at 37°C in HMI-11 medium. Where appropriate, the following drug concentrations were used: 2 μg/ml G418, 5 μg/ml hygromycin, 50 μg/ml blasticidin, and 1 μg/ml doxycycline. Transfections were performed using a Nucleofector (Amaxa) as described previously (Scahill *et al*, 2008).

### Sequencing of small 5′-triphosphate-containing RNA

Small RNA (< 200 nt) was purified from $5 \times 10^7$ cells using a combination of miRNeasy Mini (Qiagen) and RNeasy® MinElute® Cleanup (Qiagen) following the instructions of the manufacturer. Two micrograms of small RNA was treated with 1 unit of Terminator™ 5′-Phosphate-Dependent Exonuclease (TEX, Epicentre) in 1× Terminator Reaction Buffer B for 30 min at 42°C to remove 5′-monophosphate RNA. Next, the RNA was purified and one half (+5′-polyphosphatase) was treated with 20 units RNA 5′-polyphosphatase to convert 5′-triphosphate RNA to 5′-monophosphate RNA. The second half (−5′-polyphosphatase) was left untreated and served as a control. The sequencing libraries were constructed using the NEBNext® Multiplex Small RNA Library Prep Set for Illumina® (New England BioLabs) and sequenced on an Illumina® NextSeq 500.

### RNA pol II ChIP-seq

The RNA pol II ChIP was performed in cell lines, in which both alleles of *RPB9*, a subunit of the RNA pol II complex, contained an endogenous Ty1 tag. Except for minor changes, the ChIP was performed as described previously (Wedel & Siegel, 2017). In brief, $3 \times 10^8$ cells were harvested, formaldehyde-cross-linked, and permeabilized. After centrifugation, the pellet was resuspended in 600 μl of NP-S buffer and sonicated for 50 cycles at low strength in

a 15-ml tube. After centrifugation, the supernatant was transferred to a new microcentrifuge tube and 60 μl was separated as input. Immunoprecipitation of DNA bound by RBP9 was performed using Dynabeads® Protein G (Invitrogen) coupled to a BB2 antibody (Bastin *et al*, 1996) at 4°C overnight and under the presence of 300 mM NaCl. Bound material was washed, cross-links were reversed, and immunoprecipitated DNA was purified. Sequencing libraries were constructed and sequenced using an Illumina® NextSeq 500.

## Dual-luciferase assays

Luciferase activities were measured with the Dual-Glo® Luciferase Assay System from Promega. A total of $5 \times 10^6$ cells were harvested, washed with 1× phosphate-buffered saline (PBS; 2.7 nM KCl, 2 mM $KH_2PO_4$, 10 mM $Na_2HPO_4$, 137 mM NaCl), and resuspended in 200 μL 1× PBS. In a 96-well plate, 50 μl cell suspension ($1.25 \times 10^6$ cells) and 50 μl Dual-Glo® Luciferase Reagent were mixed, and after 10 min, firefly luciferase activity was measured for 1 s in a Victor Light Luminometer. Fifty microliters Dual-Glo® Stop&Glo® Reagent was added, and after 10 min, *Renilla* luciferase activity was measured. All measurements were performed at least in duplicate. To normalize for differences in cell number, firefly luciferase activity was normalized with *Renilla* luciferase activity. All raw and normalized luciferase measurements are listed in Dataset EV3.

## Antibody production, affinity purification, and characterization

Polyclonal antibodies specific for H2A.Z were raised by immunizing rabbits with the following peptide: DDAVPQAPLVGGVAMSPEQAS, following a 145-day immunization protocol (Pineda Antikörper-Service). The antisera were affinity-purified using SulfoLink Coupling Gel (ThermoFisher) with the immobilized peptide as described elsewhere (Harlow & Lane, 1999). To determine antibody specificity, different transgenic cell lines were analyzed by Western blotting (Appendix Fig S1) as described previously (Siegel *et al*, 2008).

## Overexpression of Ty1-tagged H2A.Z in *ΔH2A.Z* cells

To generate an *H2A.Z* double-knockout mutant expressing *Ty-H2A.Z* constitutively, we first introduced a Ty-tagged *H2A.Z* in a ribosomal locus using pLEW111 (Hoek *et al*, 2000) upon removal of the tetracycline repressor by digestion with BglII. The knockout of both endogenous *H2A.Z* (Tb427.07.6360) alleles was introduced by using the pyrFEKO system (Scahill *et al*, 2008). To amplify the target regions, the following primers were used: upstream of *H2A.Z* CDS: CGGTACCAACACTAGACGGC, CGTGTCCGTGTATAATGCGC; downstream of *H2A.Z* CDS: TTGTTGCCTTCAGCTCGCTA, CACTAAAACGGGCCACCTCT.

## MNase-ChIP-seq

Immunoprecipitation of mononucleosomes was performed as described previously (Wedel & Siegel, 2017). In brief, $2 \times 10^6$ cells were harvested, formaldehyde-cross-linked, and lysed using digitonin. To fragment the chromatin, the sample was digested with MNase. Immunoprecipitation of nucleosomal DNA was performed using Dynabeads® M-280 sheep anti-rabbit IgG (Invitrogen) coupled to polyclonal H3 rabbit antiserum (Gassen *et al*, 2012) at 4°C for 2 h or coupled to our custom-made polyclonal affinity-purified H2A.Z rabbit antibody (this publication) at RT for 30 min in the presence of 0.05% SDS. Bound material was washed and eluted. Cross-links were reversed, and mononucleosomal DNA was purified. ChIP-seq libraries were constructed using 35 ng of immunoprecipitated DNA or 35 ng of input DNA and sequenced on an Illumina® HiSeq 2500 or NextSeq 500.

## Mapping, normalization, and visualization of sequencing data

Immunoprecipitated DNA samples were sequenced in paired-end mode using an Illumina HiSeq 2500 or an Illumina NextSeq 500 sequencer with $2 \times 100$ and $2 \times 76$ cycles, respectively. The processing of the sequencing data was performed as described previously (Wedel & Siegel, 2017). In brief, the trimmed and clipped reads were mapped with bowtie2 version 2.1.0 (Langmead & Salzberg, 2012) or bowtie version 1.1.1 (Langmead *et al*, 2009) to the reference genomes Tb927v24 or Tb427v24 with different settings listed in Dataset EV2. The alignments in SAM format were converted to BAM format, sorted, and indexed using samtools version 0.1.19-44428 cd (Li *et al*, 2009). If necessary, non-uniquely mapped reads were removed. The genomes were downloaded from EuPathDB (Aurrecoechea *et al*, 2013) and used as reference in all analyses. To normalize the number of reads to counts per billion reads (CBR), wiggle files were generated with COVERnant version 0.3.0 subcommand ratio (available for download at GitHub https://github.com/konrad/COVERnant). For visualization of the aligned reads, coverage values for the region of interest were extracted and visualized with GraphPad Prism version 5.0b.

## Data and source code availability

All sequencing data generated for this publication have been deposited in NCBI's Gene Expression Omnibus (Edgar *et al*, 2002) and are accessible through GEO Series accession number GSE98061 (https://www.ncbi.nlm.nih.gov/geo/query/acc.cgi?acc=GSE98061). The computational data analysis was implemented as a Unix shell script, which together with further programs generated for this study is available at https://doi.org/10.5281/zenodo.438156.

**Expanded View** for this article is available online.

## Acknowledgements

We thank Stan Gorski and all members of the Siegel Laboratory for critical reading of the manuscript and Christian Janzen, Amelie Kraus, and Jens Hör for many valuable discussions. This work was funded by the Young Investigator Program of the Research Center of Infectious Diseases (ZINF) of the University of Würzburg, Germany, and the grant SI 1610/2-1 from The German Research Foundation DFG.

## Author contributions

TNS designed research; CW and RD performed research; CW, KUF, and TNS analyzed data; and CW and TNS wrote the manuscript.

## Conflict of interest

The authors declare that they have no conflict of interest.

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
