## [Review Process File · The EMBO Journal]

Manuscript EMBO-2016-95323

GT-rich promoters drive RNA pol II transcription and deposition of H2A.Z in African trypanosomes

Carolin Wedel, Konrad U. Förstner, Ramona Derr & T. Nicolai Siegel

Corresponding author: T. Nicolai Siegel, Ludwig-Maximilians-Universität München

Review timeline:

Submission date:	25 July 2016
Editorial Decision:	5 September 2016
Additional Correspondence	11 September 2016
Additional Correspondence	19 September 2016
Revision received:	10 April 2017
Editorial Decision:	22 May 2017
Revision received:	30 May 2017
Accepted:	31 May 2017

Editor: Anne Nielsen

Transaction Report:

1st Editorial Decision

5 September 2016

Thank you for submitting your manuscript for consideration by The EMBO Journal and my apologies for the slightly extended duration of the review process in this case. Your study has now been seen by two referees whose comments are shown below. As you will see, while the referees express interest in the work and topic in principle, they do not offer strong support for publication in The EMBO Journal - at least at the current stage of analysis.

I will not repeat all their individual points of criticism here, but it becomes clear that both our referees find that the depth of analysis is too limited and that the study is thus too premature for them to support its publication here. In addition, they share the more general (and potentially undermining) concern that the site of reporter integration may be a strong determinant of the effects seen and that the lower rate of H2A.Z incorporation at the reporter (relative to flanking sequences) would need to be explained. Furthermore, while the referees find it intriguing that nucleosome positions coincide with splice acceptors, they also both agree that more insight on the biological consequences of this phenomenon would have to be provided. Clearly, an extensive amount of further experimentation would be required to address these issues and to bring the study to the level of insight and significance required for publication here. Furthermore, the outcome of such experiments cannot be predicted at this point and would thus lie outside the scope and the timeframe of a revision. I therefore see little choice but to come to the conclusion that we cannot offer to publish the manuscript at this point.

Thank you in any case for the opportunity to consider this manuscript. I am sorry we cannot be more positive on this occasion, but we hope nevertheless that you will find our referees' comments

helpful.

REFEREE REPORTS

Referee #1:

In this manuscript, the Siegel laboratory addresses a longstanding question in the trypanosome field, namely how RNA polymerase II transcription of protein coding genes is initiated in this parasite and what constitutes a promoter. The authors identified GT-rich elements that can drive unidirectional transcription and promote the targeted deposition of the histone variant H2A.Z, which is the conclusion stated in the title. Although intriguing, there are several concerns that question the interpretation of the presented study.

A major concern in the experimental design is the integration site for the luciferase reporter constructs. The chosen locus is a divergent strand switch region of endogenous Pol II transcription initiation that is already occupied by H2A.Z, including in the region between the two peaks. How does integration of the reporters affect initiation for the two transcription units flanking the integration site? Comparing H2A.Z enrichment before (Fig.1B) and after integration (Fig.3A) suggests lower occupancy levels for the two large endogenous peaks. This fact could be used to argue that what the authors are actually testing in their assays is inhibition of Pol II transcription initiation, rather than stimulation. No information is presented about the relative luciferase expression levels of the most active reporters. Is it similar to rRNA promoters constructs integrated in an rDNA locus? Is it similar to reporters without promoter integrated in the same orientation within a Pol II transcription unit? Is it similar to the two transcription units flanking the integration site? If the expression levels are several-fold lower than those from an existing Pol II transcription unit, the case for the tested sequence elements acting as drivers of transcription initiation will be very weak.

Furthermore, since the reporters cannot drive transcription from regions that normally show increased H3.V levels, the title of the manuscript "GT-rich promoters drive RNA pol II transcription and deposition of H2A.Z in African trypanosomes" is an overstatement. Similar to concerns raised above, what is the explanation that GT-rich elements can only drive transcription in regions already competent for transcription?

Finally, mapping nucleosome occupancy at high resolution, the authors find that nucleosome positioning may affect RNA maturation. This is certainly possible, but not proven. The authors suggest that "small changes in nucleosome positioning may slow the polymerase at different positions, thereby affecting the choice of SAS." However, numerous nuclear run-on experiments are not supportive of this proposition.

Referee #2:

Wedel et al. study regions of transcription initiation in the parasite *Trypanosoma brucei* and find both sequence and chromatin features associated with transcription activity. Studying this parasite is relevant both because of its medical interest and because of its special gene structure, where genes are mostly transcribed in long poly-cistronic units and RNA polymerase is not regulated as in mammalian cells.

The article presents interesting observations, such as the finding of GT-rich sequences as drivers of transcription initiation and H2A.Z deposition, as well as a correlation between nucleosome depleted regions and splice acceptor sites. However, in its present state the work is somewhat preliminary and major revision is therefore recommended. Moreover, the manuscript has to be clearer in some of its explanations and interpretations of data.

Major points:

1. In both Figure 1 and 2, transcription is studied by measuring protein levels (luciferase). While this is convenient for screening purposes, a more direct transcription measurement, such as run-on or nascent RNA analysis, would considerably strengthen these arguments.

2. Interpreting the results from Figure 1 and Figure 2, the authors state that the fragments they insert confer unidirectional transcription. However, there is also transcription activity using the inverted fragments; albeit to a lesser extent. It is therefore not clear how the authors define 'unidirectional'. If the authors want to define what they observe as being unidirectional and discuss whether transcription in *T. brucei* is bidirectional or not, they should be much clearer in their definitions and explanations. Moreover, experiments should be carried out that can address this issue directly (see above).

3. In Figure 5, the authors show a correlation between nucleosome depleted regions and splice sites. While this is a very interesting observation, a demonstration of its biological relevance seems to be needed. E.g. to perform experimental manipulation of the nucleosome positioning or the splice site to check whether they functionally affect each other.

Minor points:

1. It is not clear where the ChIP-seq data in Figure 1B comes from. If it is from this work, it should be mentioned before the figure is referred to in the text. If it comes from any previous work, a reference should be added.

2. The paper would gain in clarity if it was better explained how the TSR regions for the experiments in Figure 1 were selected and how the GT-rich promoter elements were designed for the experiments in Figures 2 and 3.

3. Again for clarity, Figure 3 could show where the transcription units are in the region depicted. It would also strengthen the authors' claim to show the H2A.Z coverage in the same region without the inserted fragment.

4. Figure 3B; endogenous TSRs show a higher enrichment in H2A.Z than the inserted GT-rich fragments. How do the authors explain this?

5. Figure 4B legend; it should be clearly stated that there are panels displaying results for divergent TSRs and panels for non divergent ones.

6. Figure 3 legend; if the results from the H2A.Z ChIP come from the same experiment, they should be called the same in A and B. Using MNase-ChIP-seq data on one and ChIP-seq on the other leads to confusion.

7. Figure 5E legend; it is said that the plots are an average "across the 25% of genes containing the highest and lowest RNA levels", while in the main text, a sentence says: "the average nucleosome occupancy for the top 10% and the bottom 10% of genes". It would be interesting to see the same plots for the intermediate groups of genes. If they show the same tendency, this would strengthen the authors claims.

Additional Correspondence – authors

11 September 2016

Thank you very much for considering our manuscript for publication in the EMBO Journal.

While we appreciate that the reviewers find our study to be potentially of great interest we recognize that they raise substantial concerns regarding our findings. Nevertheless, we are confident that we can significantly extend the analysis and address all of the criticisms by performing several additional experiments within the next 3 months, especially since some of these new approaches, e.g. GRO-seq, NET-seq, are already being established in my lab.

Although I understand that your original decision stated that the current study is beyond a three-month revision, I would be very grateful for an opportunity to explain to you how we propose to extend the study to make it suitable for the EMBO Journal. Therefore, we have drafted our response to the referees' concerns and outlined how we would experimentally address them. Based on this outline, I would very much like to hear your opinion on whether the proposed experiments would make our manuscript more suitable for publication in EMBO and if you would accept to consider a revised version of the manuscript. If so we will begin the experiments immediately.

Reviewer 1

1) A major concern in the experimental design is the integration site for the luciferase reporter constructs. The chosen locus is a divergent strand switch region of endogenous Pol II transcription initiation that is already occupied by H2A.Z, including in the region between the two peaks. How does integration of the reporters affect initiation for the two transcription units flanking the integration site?

It is possible that integration of a promoter at a specific genomic locus will affect the transcription in the flanking regions. To address this concern, we will perform qPCR to measure transcript levels of genes of the regions flanking the site of promoter integration.

2) Comparing H2A.Z enrichment before (Fig.1B) and after integration (Fig.3A) suggests lower occupancy levels for the two large endogenous peaks. This fact could be used to argue that what the authors are actually testing in their assays is inhibition of Pol II transcription initiation, rather than stimulation.

The apparent drop in H2A.Z levels suggested by the differences between Figures 1B and 3A is the result of differences in data normalization and not caused by the integration of our promoter construct.

Throughout the paper we show ChIP-seq data as FPMR (fragments per million reads). This means that data is only normalized to adjust for differences in sequencing depth, i.e. if we obtain 5 million reads from one sequencing run and 2 million reads from another, we divide the number of reads from the first run by 5 and those from the second run by 2.

Such normalization does not account for the fact that some DNA sequences, like G-rich regions, are underrepresented in sequencing data. Such underrepresented regions can be easily identified by sequencing total gDNA. Indeed, when we sequenced total gDNA we noticed that the GT-rich promoter was amplified poorly and so we decided to normalize for this amplification bias. Thus, we divided the number of reads after performing the H2AZ-ChIP with the number of reads obtained from total gDNA.

This additional normalization step led to the apparent drop in H2A.Z levels. The additional normalization step was mentioned in the main text (page 7) and the figure legend, but not in the figure itself. If we normalize the data from Fig 1B the same way, the two peaks look very similar, see below. The reason the H2A.Z distribution appears smoother before promoter insertion than afterwards is due to differences in sequencing depth (20 vs 3.5 million fragments).

Number of fragments per library		
	Before insertion	After insertion
Input	21,946,536	3,590,078
ChIP	19,307,030	3,766,157

We will make this point clearer in the figure itself.

3) No information is presented about the relative luciferase expression levels of the most active reporters. Is it similar to rRNA promoters constructs integrated in an rDNA locus? Is it similar to reporters without promoter integrated in the same orientation within a Pol II transcription unit? Is it similar to the two transcription units flanking the integration site? If the expression levels are several-fold lower than those from an existing Pol II transcription unit, the case for the tested sequence elements acting as drivers of transcription initiation will be very weak.

Our data show that the GT-rich promoter leads to higher luciferase levels than the rRNA promoter (one of the most active promoters in *T. brucei*) when these promoters are inserted at the same locus between two RNA pol II TSSs.

What our data do not show is:

- the activity of the GT-rich promoter in comparison to the rRNA promoter inserted into the rDNA spacer region (a more native environment for an RNA pol I promoter)
- the activity of the GT-rich promoter in comparison to endogenous RNA pol II transcription, i.e. transcription of a gene inserted into a pol II transcription unit.

To address these points we will generate:

- a cell line carrying an rRNA promoter-driven luciferase gene in the rRNA spacer
- a cell line carrying a promoter-less luciferase gene inserted into the RNA pol II transcription unit downstream of the original site of promoter integration.

These experiments will allow us to perform detailed comparisons of the different promoter activities in the different genomic environments.

We fully agree that it will be important to show that the luciferase levels obtained from the synthetic GT-rich promoter are similar to those obtained from endogenous RNA pol II transcribed PTUs. In this regard we already have preliminary data suggesting that luciferase levels from endogenous RNA pol II transcription and from our synthetic GT-rich promoter are similar.

4) Furthermore, since the reporters cannot drive transcription from regions that normally show increased H3.V levels, the title of the manuscript "GT-rich promoters drive RNA pol II transcription and deposition of H2A.Z in African trypanosomes" is an overstatement. Similar to concerns raised above, what is the explanation that GT-rich elements can only drive transcription in regions already competent for transcription?

It is a general feature of RNA pol II promoters that their activity is affected by the local chromatin structure, i.e. there is little transcription if heterochromatin is present. This means that the activity of a promoter will depend on its genomic locus, which was nicely shown on a large scale by Akhtar et al. Thus, we were very excited to see the same phenomenon in *T. brucei* and to be able to correlate our observation to the presence of heterochromatin.

Akhtar, W., de Jong, J., Pindyurin, A. V., Pagie, L., Meuleman, W., de Ridder, J., Berns, A., Wessels, L. F., van Lohuizen, M., and van Steensel, B. (2013). Chromatin position effects assayed by thousands of reporters integrated in parallel. *Cell* 154, 914-927.

To test our assumption that GT-rich promoter-based transcription is inhibited by heterochromatin, we will do the following: Use a cell line lacking H3.V (available in our lab) and target the GT-rich promoter to the same heterochromatic regions of the genome as before (Figure EV1). Should the GT-rich promoter be able to drive luciferase transcription in this cell line, we would have identified H3.V as the repressive mark. In addition, this experiment would address the concerns regarding the choice of the site of integration.

5) Finally, mapping nucleosome occupancy at high resolution, the authors find that nucleosome positioning may affect RNA maturation. This is certainly possible, but not proven. The authors suggest that "small changes in nucleosome positioning may slow the polymerase at different positions, thereby affecting the choice of SAS." However, numerous nuclear run-on experiments are not supportive of this proposition.

Nuclear run-on assays have been used to measure rates of transcription and pausing. In addition, by adding alpha-Amanitin to nuclear run-on reactions, they can be used to identify the type of RNA polymerase (I, II or III) responsible for the observed transcription. I am not aware of studies using nuclear run-on experiments to investigate the effect of nucleosome positioning on splicing.

At the same time there are several studies showing that changes in nucleosome positioning correlate with the choice of the splice acceptor site, i.e. affect alternative splicing:

Naftelberg, S., Schor, I. E., Ast, G., and Kornblihtt, A. R. (2015). Regulation of alternative splicing through coupling with transcription and chromatin structure. *Annu Rev Biochem* 84, 165-198.

Iannone, C. et al. (2015). Relationship between nucleosome positioning and progesterone-induced alternative splicing in breast cancer cells. *RNA* 21, 360-374.

However, while a link between nucleosome positioning and splicing has been observed in many organisms, cause and consequence of this correlation is not known.

To validate our findings and to evaluate the biological significance of polyY tract length on gene expression and nucleosome positioning we will perform the following experiments:

- a) generate constructs carrying a long or a short polyY tract upstream of a luciferase gene and integrate them into the genome
- b) perform luciferase assays
- c) generate nucleosome occupancy maps to determine whether the length of polyY tracts affects nucleosome positioning

A positive correlation between polyY tract length, luciferase expression and nucleosome depletion would strengthen our hypothesis. I have previously shown that the length of polyY tracts affects gene expression in *T. brucei*.

Siegel, T. N., Tan, K. S., and Cross, G. A. M. (2005). Systematic study of sequence motifs for RNA trans splicing in *Trypanosoma brucei*. *Mol. Cell. Biol.* 25, 9586-9594.

Reviewer 2

1) In both Figure 1 and 2, transcription is studied by measuring protein levels (luciferase). While this is convenient for screening purposes, a more direct transcription measurement, such as run-on or nascent RNA analysis, would considerably strengthen these arguments.

We choose luciferase assay systems for throughput and accuracy. Small changes in expression levels can be more reliably detected by measuring luciferase levels than RNA levels. Given that we are not changing the UTRs nor the processing signals upstream or downstream of the luciferase gene, the luciferase levels should be a valid readout to measure promoter activity.

In my opinion nuclear run-on assays would only be advantageous to address the following concerns:

- a) Changes in promoter sequence affect the maturation or stability of the luciferase RNA. As a consequence, changes in luciferase levels would not be due to changes in transcription initiation but due to changes in steady state RNA levels.
- b) The luciferase gene is not transcribed by RNA pol II.

To address these concerns we will perform global nuclear run-on assays (GRO-Seq) or native elongating transcript sequencing (NET-seq). Both will address the above-mentioned possibilities and yield additional strand-specific information regarding the precise site of transcription initiation. In addition they will represent the first genome-wide data set of nascent RNA transcripts in *T. brucei* and may thus yield valuable insights about differences in the rate of transcription initiation among different PTUs or reveal sites of transcriptional pausing on a genome-wide scale.

2) Interpreting the results from Figure 1 and Figure 2, the authors state that the fragments they insert confer unidirectional transcription. However, there is also transcription activity using the inverted fragments; albeit to a lesser extent. It is therefore not clear how the authors define 'unidirectional'. If the authors want to define what they observe as being unidirectional and discuss whether transcription in *T. brucei* is bidirectional or not, they should be much clearer in their definitions and explanations. Moreover, experiments should be carried out that can address this issue directly (see above).

Our working hypothesis is that transcription initiation is bi-directional but that DNA elements flanking the site of transcription initiation lead to transcription termination in a strand-specific manner resulting in unidirectional transcription. We apologize for not being clearer and we will rewrite the respective section.

In addition, the GRO-seq or NET-seq proposed in point 2 will allow us to better identify the cause for the unidirectional transcription observed in *T. brucei* and to differentiate between the two most likely mechanisms: unidirectional transcription initiation vs bi-directional initiation followed by strand-specific termination.

3) In Figure 5, the authors show a correlation between nucleosome depleted regions and splice sites. While this is a very interesting observation, a demonstration of its biological relevance seems to be needed. E.g. to perform experimental manipulation of the nucleosome positioning or the splice site to check whether they functionally affect each other

To validate our findings and to evaluate its biological significance we will perform a series of additional experiments, see Reviewer 1 point 5.

An extensive transcriptome analysis performed by Antwi et al. identified a large number of genes that, taking into account the half-lives of their transcripts, show unexpectedly high or low expression levels. We will

determine if this unexpected regulation correlates with nucleosome positioning.

Antwi, E. B. et al. (2016). Integrative analysis of the *Trypanosoma brucei* gene expression cascade predicts differential regulation of mRNA processing and unusual control of ribosomal protein expression. *BMC Genomics* **17**, 306.

In addition, we will address all of the minor points raised by reviewer 2.

Additional Correspondence – editor

19 September 2016

I have now finally had the chance to go through your response to the referee concerns and the suggestions for additional experiments that you are planning to perform (and also discussed these with my colleagues in the editorial team). To me this sounds like a promising plan and I would be happy to look at an extensively revised version of the study once you have it ready. Of course the final outcome of the experiments is hard to predict at the present stage but if you are willing to undertake the efforts to address the referee concerns along the lines described in your response then we could consider a revised version. Since this is not an official revision there is no demand for this to be completed within a 3-month period.

Feel free to contact me with any more questions on this.

1st Revision - authors' response

10 April 2017

Response to Reviews

We thank the referees for critically reviewing and providing comments to improve our manuscript.

Please find below our responses to the issues raised by the referees.

Referee #1:

1) A major concern in the experimental design is the integration site for the luciferase reporter constructs. The chosen locus is a divergent strand switch region of endogenous Pol II transcription initiation that is already occupied by H2A.Z, including in the region between the two peaks. How does integration of the reporters affect initiation for the two transcription units flanking the integration site?

We fully agree with the reviewer that choosing the appropriate genomic environment is key when evaluating the ability of putative promoter elements to drive transcription. To avoid misleading results associated with promiscuous transcription initiation from episomes and plasmids, as reported for the closely related Kinetoplastida *Leishmania major*, we pursued a more labor-intensive route of creating stable cell lines.

To be able to evaluate promoter activity, the integration site needed to fulfill two criteria:

- a) it needed to be located in a non-transcribed region of the genome
- b) it needed to be located in a region permissive to transcription (i.e. not surrounded by heterochromatin).

Unusually for a eukaryote, almost the entire *T. brucei* genome is actively transcribed, exceptions being the small regions between two divergent transcription start sites, transcription termination sites, centromeres and subtelomeric regions. However, previous ChIP-seq experiments from our lab indicated that all but the regions between transcription start sites are enriched in histone variant H3.V (Siegel et al., 2009), a histone variant thought to mark heterochromatin in *T. brucei* (Reynolds et al., 2016; Schulz et al., 2016).

Thus, the region between two divergent transcription start sites is the best and possibly only option to carry out the proposed assays. The importance of genomic context and the suitability of the site of integration we chose is underlined by our finding that integration of the same reporter construct into a heterochromatic region did not lead to transcription.

To determine whether the integration of our promoter constructs affected transcription of the flanking genes, a very valid concern, we have measured

Figure EV3 – Impact of GT-rich element insertion on the transcription of the flanking PTUs.

Transcript levels of genes flanking the site of promoter insertion were determined by qPCR after insertion of the GT_416_nt element and after insertion of the same constructs lacking a promoter element. Transcript levels after insertion of the promoter-less construct were set to 1 (grey bars) and the relative fold-change in transcript levels after insertion of the GT_416_nt element is shown as green bars. Measurements were performed in triplicates and for two independent clones, shown are averages \pm SD, for details see Appendix and Dataset EV4. The black arrow marks the site of reporter insertion.

transcript levels of four genes flanking the site of integration and found no significant change. The qPCR result is shown in Figure EV3.

2) Comparing H2A.Z enrichment before (Fig.1B) and after integration (Fig.3A) suggests lower occupancy levels for the two large endogenous peaks. This fact could be used to argue that what the authors are actually testing in their assays is inhibition of Pol II transcription initiation, rather than stimulation.

We apologize for not having been clearer about the differences between these figures:

The apparent drop in H2A.Z levels suggested by the differences between Figures 1B and 3A (original version of the manuscript) was the result of differences in data normalization and not caused by the integration of our promoter construct.

Throughout the paper we showed ChIP-seq data as FPMR (fragments per million reads). This means that data were only normalized to adjust for differences in sequencing depth, i.e. if we obtained 5 million reads from one sequencing run and 2 million reads from another, we divided the number of reads from the first run by 5 and those from the second run by 2.

Such normalization does not account for the fact that some DNA sequences, such as G-rich regions, are typically underrepresented in sequencing data. Such underrepresented regions can be easily identified by sequencing total gDNA (e.g. input material). Indeed, when we sequenced the input material we noticed that the GT-rich promoter was amplified poorly and so we decided to normalize for this amplification bias. Thus, we divided the number of reads after performing the H2AZ-ChIP with the number of reads obtained from total gDNA.

This additional normalization step led to the apparent drop in H2A.Z levels. The additional normalization step was mentioned in the main text (page 7) and the figure legend, but not in the figure itself. When we normalized the data from Fig 1B the same way, the two peaks look very similar, see below. The reason the

H2A.Z distribution appears smoother before promoter insertion than afterwards is due to differences in sequencing depth (20 vs 3.5 million fragments).

Number of fragments per library

	Before insertion	After insertion
Input	21,946,536	3,590,078
ChIP	19,307,030	3,766,157

In the revised manuscript, we have changed this section significantly and have removed the figure shown above to avoid confusion. Instead we now show H2A.Z levels across the GT-rich sequence and *FLUC* relative to the adjacent endogenous H2A.Z-rich region. In addition, we have repeated the H2A.Z ChIP-seq experiments (8 days post infection) (Fig 3D).

3) No information is presented about the relative luciferase expression levels of the most active reporters. Is it similar to rRNA promoters constructs integrated in an rDNA locus? Is it similar to reporters without promoter integrated in the same orientation within a Pol II transcription unit? Is it similar to the two transcription units flanking the integration site? If the expression levels are several-fold lower than those from an existing Pol II transcription unit, the case for the tested sequence elements acting as drivers of transcription initiation will be very weak.

It is generally assumed that RNA pol II promoter motifs are absent in *T. brucei* and that an open chromatin structure is sufficient for RNA pol II transcription initiation. The goal of this study was to determine the importance of DNA sequence elements in transcription initiation, histone variant recruitment, and nucleosome positioning.

Our data show that the GT-rich promoter leads to 18.5-fold higher luciferase levels than the same construct lacking the GT-rich promoter element

(Fig 3C). Thus, our study for the first time demonstrates the importance of DNA sequence elements in RNA pol II transcription in *T. brucei*. However, we do not mean to claim that DNA sequence motifs are the only factor important for transcription initiation.

What we had not shown were the levels of RNA pol I and RNA pol II transcription at endogenous sites.

- a) To determine the activity of an endogenous RNA pol II promoter, we generated a cell line carrying a luciferase gene inserted into an endogenous RNA pol II transcription unit. For this cell line we find the luciferase activity to be 11.6-fold higher compared to the cell line containing a GT-rich promoter-driven luciferase (EV4A). Nevertheless, the GT-rich element-induced levels are 18.5-fold higher than those from our negative control lacking a promoter and inserted between two divergent transcription start sites. Thus, while the GT-rich element inserted between two divergent transcription start sites is clearly capable of inducing transcription initiation, demonstrating the importance of DNA sequence elements in transcription, the observed luciferase levels are lower than those obtained from endogenous sites. Based on our finding that the site of integration strongly affects transcriptional activity, we hypothesize that 'features' specific to the different target sites, e.g. differences in chromatin composition, contribute to observed differences in luciferase activity.
- b) To determine the activity of an endogenous RNA pol I promoter we generated a cell line carrying an rRNA promoter-driven luciferase gene in the rRNA spacer. *T. brucei* contains rRNA spacers on different chromosomes. Previously, it was reported that rRNA activity can vary depending on the rRNA spacer and so we created four different cell lines. As expected we find luciferase activity for these cell lines to be very high, 38.7-193.6-fold higher than for the rRNA promoter inserted between divergent transcription start sites. Given that RNA pol I is not equally distributed across the nucleus but strongly enriched within the nucleolus, where the rDNA spacer is found, the strong differences in transcriptional activity are in line with our expectations.

4) Furthermore, since the reporters cannot drive transcription from regions that normally show increased H3.V levels, the title of the manuscript "GT-rich promoters drive RNA pol II transcription and deposition of H2A.Z in African trypanosomes" is an overstatement. Similar to concerns raised above, what is the explanation that GT-rich elements can only drive transcription in regions already competent for transcription?

It is a common feature of the RNA pol II promoter activity to be affected by the local chromatin structure, i.e. there is little transcription if heterochromatin is present. This means that the activity of a promoter will depend on its genomic

context, which was nicely shown on a genomic scale by Akhtar et al. (2013). Thus, we were very excited to see the same phenomenon in *T. brucei* and to be able to correlate our observation with the presence of heterochromatin.

Akhtar, W., de Jong, J., Pindyurin, A. V., Pagie, L., Meuleman, W., de Ridder, J., Berns, A., Wessels, L. F., van Lohuizen, M., and van Steensel, B. (2013). Chromatin position effects assayed by thousands of reporters integrated in parallel. *Cell* 154, 914-927.

To test our assumption that GT-rich promoter-based transcription is inhibited by H3.V-mediated heterochromatin, we generated a Δ H3.V cell line and targeted the GT-rich promoter to the same heterochromatic regions of the genome as before (Figure EV1C). However, even in the absence of H3.V the GT-rich promoter was unable to drive luciferase transcription. Thus, other chromatin marks like the histone variant H4.V, 3D genome organization or additional sequence elements may be important contributors to transcription as well. Previously we had shown that both H3.V and H4.V mark sites of transcription termination (Siegel et al., 2009). However, for technical reasons we were not able to generate a cell line lacking both alleles of H3.V and both alleles of H4.V that could be used for our promoter assays.

Nevertheless, the active transcription of luciferase after insertion of the GT-rich promoter at divergent TSRs is a clear indication that this element is capable of serving as a promoter element. To address the concerns raised by the reviewer and to avoid any overstatement, we have changed the title to:

GT-rich promoters can drive RNA pol II transcription and deposition of H2A.Z in African trypanosomes

5) Finally, mapping nucleosome occupancy at high resolution, the authors find that nucleosome positioning may affect RNA maturation. This is certainly possible, but not proven. The authors suggest that "small changes in nucleosome positioning may slow the polymerase at different positions, thereby affecting the choice of SAS." However, numerous nuclear run-on experiments are not supportive of this proposition.

While alternative splicing is wide-spread in *T. brucei* (Nilsson et al., 2010) nothing is known about the factors contributing to the choice of splice acceptor sites in this parasite. The hypothesis that changes in nucleosome positioning can affect the choice of splice acceptor sites is based on three general observations made in different organisms:

1) The tight association of DNA with histones to form nucleosomes slows polymerase elongation:

Allfrey, V. G., Littau, V. C., and Mirsky, A. E. (1963). On the role of histones in regulation ribonucleic acid synthesis in the cell nucleus. Proc. Natl. Acad. Sci. USA 49, 414-421.

2) The speed of RNA pol II elongation can affect splice acceptor site choice:

de la Mata, M., Alonso, C. R., Kadener, S., Fededa, J. P., Blaustein, M., Pelisch, F., Cramer, P., Bentley, D., and Kornblihtt, A. R. (2003). A slow RNA polymerase II affects alternative splicing *in vivo*. Mol Cell 12, 525-532.

Dujardin, G., Lafaille, C., de la Mata, M., Marasco, L. E., Muñoz, M. J., Le Jossic-Corcós, C., Corcos, L., and Kornblihtt, A. R. (2014). How slow RNA polymerase II elongation favors alternative exon skipping. Mol Cell 54, 683-690.

3) Changes in nucleosome positioning correlate with changes in the choice of the splice acceptor site, i.e. affect alternative splicing:

Naftelberg, S., Schor, I. E., Ast, G., and Kornblihtt, A. R. (2015). Regulation of alternative splicing through coupling with transcription and chromatin structure. Annu Rev Biochem 84, 165-198.

Iannone, C. et al. (2015). Relationship between nucleosome positioning and progesterone-induced alternative splicing in breast cancer cells. RNA 21, 360-374.

Given the importance of nucleosome positioning in gene expression across eukaryotes, we decided to investigate the role of nucleosome positioning in gene regulation in *T. brucei*. To this end the following data sets were generated (for the first time in *T. brucei*):

- 1) Multiple high-resolution nucleosome occupancy maps.
- 2) A map of RNA pol II enrichment across the *T. brucei* genome. A common proxy for polymerase elongation speed is polymerase enrichment, as a slowdown of polymerase elongation leads to higher polymerase levels.

Analysis of the newly generated data revealed that RNA pol II occupancy mirrors nucleosome occupancy suggesting that nucleosomes affect RNA pol II elongation speed *in vivo*. In addition, we find that highly expressed genes are preceded by a much stronger drop in nucleosome and RNA pol II occupancy than weakly expressed genes. Since in *T. brucei* genes are organized in long polycistronic transcription units, which leave very little room for gene regulation at the level of transcription initiation, we hypothesize that the slowdown in RNA pol II elongation positively affects *trans*-splicing efficiency thereby leading to increased levels of mature RNA. In addition, it is possible that life cycle specific patterns of nucleosome occupancy will slow RNA pol II at different positions, resulting in the life cycle specific alternative splicing events that have been observed in *T. brucei*.

Referee #2:

Major points:

1) In both Figure 1 and 2, transcription is studied by measuring protein levels (luciferase). While this is convenient for screening purposes, a more direct transcription measurement, such as run-on or nascent RNA analysis, would considerably strengthen these arguments.

We chose luciferase assay systems for throughput and accuracy. Small changes in expression levels can be very reliably detected by measuring luciferase levels without having to rely on radioactively labeled RNA. Given that we are not changing the UTRs nor the processing signals upstream or downstream of the luciferase gene, the luciferase levels should be a valid readout to measure promoter activity.

In addition, our luciferase reporter system allowed us to assay promoter activity without affecting native nuclear organization, e.g. chromatin structure or polymerase compartmentalization.

In our opinion nuclear run-on assays would nevertheless be advantageous to determine the precise sites of transcription initiation and, in combination with alpha-amanitin treatment, the polymerase responsible for the observed transcription.

To address these points on a genome-wide scale, we established a protocol to precisely map sites of transcription in a strand-specific manner. To reduce the amount of highly abundant processed transcripts such as rRNA, we enriched the sample for short RNA transcripts and treated it with a 5'-monophosphate-dependent exonuclease (TEX, Epicentre). Using the short RNA containing reduced levels of processed transcripts we generated two libraries:

- 1) one library from transcripts carrying a 5'-triphosphate (unprocessed primary transcripts) and from transcripts carrying a 5'-monophosphate (processed RNA, e.g. tRNA)
- 2) one library from transcripts only carrying a 5'-monophosphate (not containing primary transcripts)

A comparison of these two libraries allowed us to identify primary transcripts, i.e. sites of transcription initiation.

In addition to the map of transcription start sites we generated the first RNA pol II ChIP-seq data for *T. brucei*. Combining the two datasets allowed us to unequivocally identify and characterize sites of RNA pol II transcription initiation.

Thus we used our genome-wide data sets to identify putative promoter elements and the luciferase reporter system to study the effect of the identified sequence motifs.

2) Interpreting the results from Figure 1 and Figure 2, the authors state that the fragments they insert confer unidirectional transcription. However, there is also transcription activity using the inverted fragments; albeit to a lesser extent. It is therefore not clear how the authors define 'unidirectional'. If the authors want to define what they observe as being unidirectional and discuss whether transcription in *T. brucei* is bidirectional or not, they should be much clearer in their definitions and explanations. Moreover, experiments should be carried out that can address this issue directly (see above).

We apologize for not being clearer. After having generated strand-specific TSS mapping data, we find the ratio between endogenous primary sense and antisense transcripts to be 4:1. Thus, while there is a clear directionality in transcription initiation from endogenous sites, we also detected antisense transcription.

The results obtained with the two transcription start regions tested in our luciferase assays are very similar to the ratio measured from endogenous loci, we observed 4-fold more transcription in one direction than in the opposite direction. However, since we do not have any information on what causes the directionality, we have removed any speculation in this regard from the discussion.

3) In Figure 5, the authors show a correlation between nucleosome depleted regions and splice sites. While this is a very interesting observation, a demonstration of its biological relevance seems to be needed. E.g. to perform experimental manipulation of the nucleosome positioning or the splice site to check whether they functionally affect each other.

What we had observed was that splice sites are depleted of nucleosomes and that this depletion is more pronounced for highly expressed genes. Our hypothesis was that strongly positioned nucleosomes flanking these depleted regions may function as 'speed bumps' to slow down RNA pol II elongation, thereby increasing splicing efficiency.

This hypothesis is supported by our newly generated RNA pol II ChIP-seq data. We see RNA pol II levels to increase at exon boundary, pointing to a slowdown in elongation (Fig 5D).

In addition, as suggested by the reviewer, we performed experimental manipulations and generated an additional set of three high-resolution nucleosome occupancy maps to study cause and consequence of the polyY tract in nucleosome positioning and *trans*-splicing.

The importance of the polyY tract for *trans*-splicing has been shown using transient transfection assays and in vitro but never in a genomic context where nucleosome occupancy may play a role as well.

Thus we generated three cell lines carrying a) a polyY of the highly

expressed GPEET gene, b) a highly T-rich polyY tract and c) no polyY tract upstream of the luciferase reporter. Next we measured luciferase levels in these three cell lines and generated high-resolution nucleosome occupancy maps. The luciferase assays confirmed the importance of the polyY tract for efficient processing, as the cell line lacking a polyY tract yielded no luciferase activity (Fig 6A).

The new nucleosome occupancy maps were interesting in several aspects. As expected, the homopolymeric nature of the highly T-rich polyY tract, giving rise to a rather rigid DNA double helix, resulted in an expansion of the nucleosome-depleted region (Fig 6B, middle panel). Importantly however, the cell line lacking a polyY tract still contained a well-defined nucleosome-depleted region upstream of the luciferase ORF (Fig 6B, lower panel). Thus, while the polyY is important for efficient *trans*-splicing and can lead to nucleosome depletion (if it contains homopolymeric stretches of Ts), there must be other elements that ensure the presence of a nucleosome-depleted region. Such elements may be located in the 5'UTR.

Thus, unlike most other studies that have investigated the link between nucleosome positioning and splicing purely based on correlative data, we demonstrated the effect of experimental manipulation on nucleosome positioning.

Minor points:

1) It is not clear where the ChIP-seq data in Figure 1B comes from. If it is from this work, it should be mentioned before the figure is referred to in the text. If it comes from any previous work, a reference should be added.

The data in the Figure 1B (original version of the manuscript) was from a previous ChIP-seq experiment (Siegel et al., 2009). In this revised version of the manuscript we are using only H2A.Z MNase ChIP-seq data generated in this study. The data source files for all figures are listed in Dataset EV2.

2) The paper would gain in clarity if it was better explained how the TSR regions for the experiments in Figure 1 were selected and how the GT-rich promoter elements were designed for the experiments in Figures 2 and 3.

The two TSRs analyzed in our luciferase assay are representative of other TSRs found in the *T. brucei* genome. To avoid partial translocations of genes, which could lead to secondary effects, we selected TSRs that did not contain genes spanning their boundaries. In addition we chose TSRs lacking NotI and XhoI restriction sites since these sites were required for plasmid linearization prior to transfection.

The synthetic GT promoters (GT_210_nt, GT_206_nt, GT_416_nt) were synthesized by Integrated DNA Technologies (IDT) and are composed of 10mer motifs enriched in the coding strand (listed in Dataset EV1). The 10mers were

ordered so that they met synthesis requirements set by IDT and, where necessary, As and Cs were inserted between 10mers to reduce the GT content and to allow synthesis.

We have added this information to the appendix.

3) Again for clarity, Figure 3 could show where the transcription units are in the region depicted. It would also strengthen the authors' claim to show the H2A.Z coverage in the same region without the inserted fragment.

We have changed this section significantly and have removed the figure referred to by the reviewer. Instead we now show H2A.Z levels across the GT-rich sequence and *FLUC* relative to the adjacent endogenous H2A.Z rich region (Fig 3D, see also point 2 by reviewer 1). In addition, we have repeated the H2A.Z ChIP-seq experiments (8 days post infection) (Fig 3D).

7) Figure 3B; endogenous TSRs show a higher enrichment in H2A.Z than the inserted GT-rich fragments. How do the authors explain this?

The nucleosome occupancy maps generated from the three cell lines carrying the GT-rich promoter element (Fig 6B) revealed that the GT-rich sequence is strongly depleted of nucleosomes. We suspect this depletion to be caused by the long homopolymeric G and T stretches present in the promoter element. Given the low overall amount of nucleosomes in this region, H2A.Z levels cannot reach the levels of endogenous TSRs.

8) Figure 4B legend; it should be clearly stated that there are panels displaying results for divergent TSRs and panels for non divergent ones.

We have added this information to the legend. The legend of Fig 4B now reads:

The relative enrichment of H2A.Z and total nucleosome occupancy averaged across all divergent TSRs (left panel) and non-divergent TSRs (right panel) are plotted relative to the TSR center. H2A.Z, window size: 101 bp, step size: 101 bp; mononucleosomes, window size: 11 bp, step size: 11 bp.

9) Figure 3 legend; if the results from the H2A.Z ChIP come from the same experiment, they should be called the same in A and B. Using MNase-ChIP-seq data on one and ChIP-seq on the other leads to confusion.

We are sorry about the confusion. All data from this figure came from MNase-ChIP-seq experiments. For the revised version of this manuscript all ChIP assays were done according to our MNase-ChIP-seq protocol. The only exception being the RNA pol II ChIP-seq for which we used a more traditional sonication-based protocol as this led to higher immunoprecipitation efficiencies.

10) Figure 5E legend; it is said that the plots are an average "across the 25% of genes containing the highest and lowest RNA levels", while in the main text, a sentence says: "the average nucleosome occupancy for the top 10% and the bottom 10% of genes". It would be interesting to see the same plots for the intermediate groups of genes. If they show the same tendency, this would strengthen the authors claims.

As described in the figure legend, in the main text it should have read: "the average nucleosome occupancy for the top 25% and the bottom 25% of genes". We have corrected this mistake.

The nucleosome and RNA pol II levels for the intermediate data set (middle 25%) look similar to that of the highly expressed genes but the drop upstream of the SAS is a little bit less pronounced, see figure below and Fig 5D.

Thank you for submitting a revised version of your manuscript. It has now been seen by two of the original referees whose comments are shown below. As you will see they both find that all major criticisms have been sufficiently addressed and they recommend the manuscript for publication in The EMBO Journal, pending clarification of a few minor points. I would therefore invite you to submit a final revision of the manuscript in which you address the remaining concerns from the referees as well as the following editorial points concerning text and figure:

-> Please provide the manuscript as a .doc file

-> We generally require that all information relevant to the main experiments in the manuscript should be included in Materials and Methods. I would therefore ask you to move the following sections from the supplemental materials to the main manuscript file: RNA pol II ChIP-seq, Antibody production, affinity purification and characterization, Generation of TbH2A.Z-/-, Mapping, normalization and visualization of sequencing data

-> The GEO number given for your sequencing data leads to an entry that has been deleted. Could you please check that the correct number is included in the manuscript and the checklist?

-> We can accommodate up to five typeset Expanded View figures per paper published in The EMBO Journal and I noticed that your manuscript currently has seven. Could you please move two of them to the Appendix file? This involves relabeling the figures Appendix figure S1, S2 etc and updating the callouts in the manuscript text accordingly. Please see our author guidelines for more detail on this <http://emboj.embopress.org/authorguide>

-> Please include a Table of Contents on the first page of the Appendix file

-> For the EV tables and EV datasets the corresponding legends should be included in a separate tab in the .xls sheet rather than listed as part of the main manuscript.

-> Please ensure that the number of replicas used for calculating statistics is indicated in all relevant figure legends (figs 3+6, EV figs 1, 3, 4, and 6)

-> Papers published in The EMBO Journal include a 'Synopsis' to further enhance discoverability. Synopses are displayed on the html version of the paper and are freely accessible to all readers. The synopsis includes a short standfirst - written by the handling editor - as well as 2-5 one sentence bullet points that summarise the paper and are provided by the authors. I would therefore ask you to include your suggestions for bullet points.

-> In addition, I would encourage you to provide an image for the synopsis. This image should provide a rapid overview of the question addressed in the study but still needs to be kept fairly modest since the image size cannot exceed 550x400 pixels.

Thank you again for giving us the chance to consider your manuscript for The EMBO Journal, I look forward to receiving your final revision.

 REFEREE REPORTS

Referee #1:

In this revised manuscript the authors did an excellent job addressing the reviewer comments and in my opinion have erased the concerns raised in the initial review. In addition, they included new experimental data that clearly solidify the interpretations and conclusions in the manuscript.

The only very minor point is that the following text should be modified to avoid confusion with relating expressed genes to promoter strength. Transcript levels would be more appropriate.

"In addition, when grouping genes in highly (top 25%), intermediately (middle 25 %) and weakly expressed genes (bottom 25%) based on RNA levels, we find a well defined NDR upstream of highly and intermediately expressed genes but not upstream of weakly expressed genes (Fig 5D)."

Referee #2:

The authors have done a good job in addressing the points raised by the reviewers and the manuscript has clearly improved. A few issues need to be corrected before publication:

Major comment:

1. With the new datasets, the authors estimate that transcription initiates sense/antisense with a 4:1 ratio. While this indicates a preference for one direction, it also shows that there is bidirectional transcription. The authors keep talking about unidirectional transcription. They should either omit this from the paper or clearly define what they mean by unidirectional transcription when it is predominantly, but certainly not exclusively, in one direction.

Minor comments:

1. The ChIP-seq data for H2A.Z are already shown in Figure 1, but only introduced later in the manuscript. It should be mentioned when used for the first time.
2. In Figure 3D, the legend on the panel states "21/97 days post transfection". I cannot find a description in neither figure legend nor text.
3. In Figure 4B, the x-axis is labeled as "distance from TSR center (bp)", but in the panels for divergent TSRs, there are two zeros, while there should only be one center. How should this be understood? It is different from Figure 1C.
4. The authors say on page 9 that they averaged the nucleosome occupancy using the ATG of the first gene because they did not have "sharp peaks of transcription initiation", but with the new datasets they obtained, they have data for transcription initiation and could average the nucleosomes based on that.
5. On page 10, the authors introduce Figure 6: "to better understand cause and consequence of nucleosome..." If I understand it correctly, the panels in Figure 6B (especially the middle and the lower ones) are not so different and therefore their conclusion is that the polyY tract can affect nucleosome positioning, but there must be other elements that also contribute. Thus, it seems that cause and consequence cannot clearly be distinguished.

2nd Revision - authors' response

30 May 2017

Response to editorial points

-> Please provide the manuscript as a .doc file

We have uploaded the manuscript as .doc file.

-> We generally require that all information relevant to the main experiments in the manuscript should be included in Materials and Methods. I would therefore ask you to move the following sections from the supplemental materials to the main manuscript file: RNA pol II ChIP-seq, Antibody production, affinity purification and characterization, Generation of TbH2A.Z-/-, Mapping, normalization and visualization of sequencing data

As required by EMBO, we have moved the sections listed above to the main manuscript.

-> The GEO number given for your sequencing data leads to an entry that has been deleted. Could you please check that the correct number is included in the manuscript and the checklist?

We have replaced the old GEO number with a new functional one.

-> We can accommodate up to five typeset Expanded View figures per paper published in The EMBO Journal and I noticed that your manuscript currently has seven. Could you please move two of them to the Appendix file? This involves relabeling the figures Addendix figure S1, S2 etc and updating the callouts in the manuscript text accordingly. Please see our author guidelines for more detail on this <http://emboj.embopress.org/authorguide>

We have moved the Figure EV2 and Figure EV5 to the appendix and renamed the other Expanded View Figures accordingly.

-> Please include a Table of Contents on the first page of the Appendix file

We have added a table of contents for the Appendix file.

-> For the EV tables and EV datasets the corresponding legends should be included in a separate tab in the .xls sheet rather than listed as part of the main manuscript.

The EV tables already contained the figure legend, thus we simply removed the legends from the main manuscript.

For the EV datasets we have included a separate tab with the figure legends and removed the legends from the main manuscript.

-> Please ensure that the number of replicas used for calculating statistics is indicated in all relevant figure legends (figs 3+6, EV figs 1, 3, 4, and 6)

We have included the relevant numbers.

-> Papers published in The EMBO Journal include a 'Synopsis' to further enhance discoverability. Synopses are displayed on the html version of the paper and are freely accessible to all readers. The synopsis includes a short standfirst - written by the handling editor - as well as 2-5 one sentence bullet points that summarise the paper and are provided by the authors. I would therefore ask you to include your suggestions for bullet points.

- RNA pol II transcription initiates across a 2kb-wide region upstream of polycistronic transcription units.
- GT-rich sequence elements enriched at transcription start sites can induce transcription and recruit the histone variant H2A.Z.
- Sites enriched in H2A.Z show increased sensitivity to MNase.
- Nucleosome occupancy upstream of genes correlates with RNA pol II enrichment and transcript levels.
- Composition of polyY tract affects nucleosome positioning and transcript levels.

-> In addition, I would encourage you to provide an image for the synopsis. This image should provide a rapid overview of the question addressed in the study but still needs to be kept fairly modest since the image size cannot exceed 550x400 pixels.

We included an image for the synopsis.

Response to Reviews

We are happy to read that our additional experiments were able to address all major concerns and thank the referees for their help in improving our manuscript. Please find below our responses to the issues raised by the referees:

Referee #1:

1) The only very minor point is that the following text should be modified to avoid confusion with relating expressed genes to promoter strength. Transcript levels would be more appropriate.

"In addition, when grouping genes in highly (top 25%), intermediately (middle 25%) and weakly expressed genes (bottom 25%) based on RNA levels, we find a well defined NDR upstream of highly and intermediately expressed genes but not upstream of weakly expressed genes (Fig 5D)."

As suggested, we have replaced high, intermediate and weakly expressed genes with high, intermediate and low transcript levels. This can be found on pages 9 and 10.

Referee #2:

Major comment:

1) With the new datasets, the authors estimate that transcription initiates sense/antisense with a 4:1 ratio. While this indicates a preference for one direction, it also shows that there is bidirectional transcription. The authors keep talking about unidirectional transcription. They should either omit this from the paper or clearly define what they mean by unidirectional transcription when it is predominantly, but certainly not exclusively, in one direction.

We have removed all references to unidirectional and strand-specific transcription as this is indeed misleading. Where relevant, we mention that we see a strand bias in transcription initiation or refer to directional transcription, which we define as more sense transcription than antisense transcription (page 6).

Minor comments:

1. The ChIP-seq data for H2A.Z are already shown in Figure 1, but only introduced later in the manuscript. It should be mentioned when used for the first

time.

We now mention the H2A.Z MNase-ChIP-seq data when it is first used, on page 5.

2. In Figure 3D, the legend on the panel states "21/97 days post transfection". I cannot find a description in neither figure legend nor text.

We apologise for the missing information and added the explanation to the figure legend.

3. In Figure 4B, the x-axis is labeled as "distance from TSR center (bp)", but in the panels for divergent TSRs, there are two zeros, while there should only be one center. How should this be understood? It is different from Figure 1C.

We had shown two zeros on the x-axis of divergent TSR because they contain two sites of transcription initiation, illustrated in 4a. However, as this is confusing to the reader, we have relabeled the x-axis in Fig 1A, 1C and 4B.

For non-divergent TSRs we now label the x-axis:
Distance from midpoint of non-divergent TSRs [bp]

For divergent TSRs we now label the x-axis:
Distance from midpoint of regions between divergent TSRs [bp]

4. The authors say on page 9 that they averaged the nucleosome occupancy using the ATG of the first gene because they did not have "sharp peaks of transcription initiation", but with the new datasets they obtained, they have data for transcription initiation and could average the nucleosomes based on that.

While the new datasets allowed us to narrow down the region of transcription initiation to around 2000 bp, nucleosomes are positioned on average every ~160-180 nt. Thus, even though we have more precise data regarding the sites of transcription initiation, the fact that transcription initiates over a broad region makes it difficult to choose a defined reference point that is needed for the average plots.

5. On page 10, the authors introduce Figure 6: "to better understand cause and consequence of nucleosome..." If I understand it correctly, the panels in Figure 6B (especially the middle and the lower ones) are not so different and therefore their conclusion is that the polyY tract can affect nucleosome positioning, but there must be other elements that also contribute. Thus, it seems that cause and consequence cannot clearly be distinguished.

This is a good point. While it was our goal to learn something about cause and consequence of NDRs, we never removed the NDR, thus we cannot say anything about the consequence of not having one. The fact that removal of the polyY tract did not lead to a loss of the NDR, tells us that something else is responsible for the establishment of the NDR. Thus, we learned something about the cause. We have modified the statement in the manuscript accordingly on page 10.

Thank you for sending the final revision of your study, I am pleased to inform you that the manuscript has now been officially accepted for publication in the EMBO Journal.

Thank you for your contribution to The EMBO Journal and congratulations on this nicely executed work!

Corresponding Author Name: Tim Nicolai Siegel

Manuscript Number: EMBOJ-2017-97108